# Glucose promotes cell growth by suppressing branched-chain amino acid degradation

Dan Shao[1], Outi Villet[1], Zhen Zhang[1], Sung Won Choi[1], Jie Yan[2], Julia Ritterhoff[1], Haiwei Gu[1], Danijel Djukovic[1], Danos Christodoulou[2], Stephen C. Kolwicz Jr[1], Daniel Raftery [1,3] & Rong Tian[1]

Glucose and branched-chain amino acids (BCAAs) are essential nutrients and key determinants of cell growth and stress responses. High BCAA level inhibits glucose metabolism but reciprocal regulation of BCAA metabolism by glucose has not been demonstrated. Here we show that glucose suppresses BCAA catabolism in cardiomyocytes to promote hypertrophic response. High glucose inhibits CREB stimulated KLF15 transcription resulting in downregulation of enzymes in the BCAA catabolism pathway. Accumulation of BCAA through the glucose-KLF15-BCAA degradation axis is required for the activation of mTOR signaling during the hypertrophic growth of cardiomyocytes. Restoration of KLF15 prevents cardiac hypertrophy in response to pressure overload in wildtype mice but not in mutant mice deficient of BCAA degradation gene. Thus, regulation of KLF15 transcription by glucose is critical for the glucose-BCAA circuit which controls a cascade of obligatory metabolic responses previously unrecognized for cell growth.

[1] Department of Anesthesiology and Pain Medicine, Mitochondria and Metabolism Center, University of Washington, Seattle, WA 98109, USA. [2] Department of Medicine, NMR Laboratory of Physiological Chemistry, Brigham and Women's Hospital, Boston, MA 02115, USA. [3] Fred Hutchinson Cancer Research Center, 1100 Fairview Ave, Seattle, WA 98109, USA. Correspondence and requests for materials should be addressed to R.T. (email: rongtian@u.washington.edu)

Cell proliferation, such as during fetal development or cancerous growth, is highly dependent on glucose metabolism which not only provides a source of energy but also supplies metabolites for biosynthesis of membrane lipids and nucleic acids, as well as contributes to epigenetic and post-translational modifications[1–4]. The growth of post-mitotic cells, i. e, increases of cell mass, is also associated with increased glucose utilization[5–8]. The hypertrophic growth of the heart under stress conditions is a hallmark of pathological remodeling which ultimately leads to heart failure[9,10]. Cardiac pathological hypertrophy is accompanied by a switch of substrate metabolism from fatty acid oxidation to glucose use, resulting in a fetal like metabolic profile[5,11,12]. While such a metabolic reprogramming has been shown maladaptive for sustaining energy supply[13,14], its role in cardiomyocyte growth is poorly understood. It has been suggested that metabolites of glucose, in particular, glucose-6-phosphate, play an important role in cell growth by activating the mechanistic target of rapamycin (mTOR) complex[15,16]. However, enhancing glucose utilization alone does not drive cell growth suggesting that it is unlikely that glucose or its metabolites directly stimulates mTOR[17,18].

The Krüppel-like factors (KLFs) are a family of gene regulatory proteins that govern the transcriptional regulation of cell proliferation, differentiation and metabolism[19,20]. KLF15 has been shown as a negative regulator of cardiomyocyte hypertrophy, and KLF15 expression is negatively correlated with cell growth in proliferating and non-proliferating cells[21,22]. Despite the close association, molecular mediator(s) connecting the downstream of KLF15 and the growth signals have been elusive. The driver or suppressor of KLF15 expression at different developmental stages is also unknown. Interestingly, the expression profile of KLF15 during developmental and postnatal phases predicts a negative relationship with glucose utilization[23,24].

Using an unbiased transcriptome analysis, we here identified glucose as a negative regulator of KLF15 expression. Increases of intracellular glucose, via downregulation of KLF15, suppressed the expression of BCAA degradation enzymes resulting in BCAA accumulation. While glucose did not directly modify the upstream signaling of cell growth, e.g., phosphatidylinositide-3-kinase (PI3K) or mitogen-activated protein kinase (MAPK), activation of glucose-KLF15-BCAAs axis was required for the signal transduction through the mTOR complex. Furthermore, upregulation of KLF15 in an in vivo model of cardiac pressure overload prevented the development of cardiomyocyte hypertrophy in wildtype mice but not in mice deficient of BCAA degradation genes. Collectively, our results identified a regulatory circuit between glucose and BCAA, and unveiled a previously unrecognized pathway for metabolic regulation of cell growth.

## Results

**Glucose negatively regulates BCAA degradation.** Previously we have demonstrated that overexpression of an insulin independent glucose transporter *Glut1* in cardiomyocytes (Glut1-TG) increased glucose uptake and utilization but did not alter cardiac function or survival of the mice under unstressed condition[17,25]. RNA microarray and gene ontology analysis of the Glut1-TG hearts revealed that downregulation of the "branched-chain amino acids (BCAAs) degradation" pathway was among the most enriched terms (Fig. 1a), which was confirmed by the Kyoto Encyclopedia of Genes and Genomes (KEGG) pathways analysis (Supplementary Table 1). A total of 26 out of 46 genes in the BCAA degradation pathway was downregulated in the Glut1-TG heart (Supplementary Fig. 1a, b), and the changes of the key enzymes were confirmed by real-time PCR (Fig. 1b, c)[26]. The mRNA levels of branched-chain amino acid transaminase 2

(BCAT2) and branched-chain alpha-keto acid dehydrogenase (BCKDH) complex, two enzymes that catalyzed the initial and committed steps of BCAA degradation to branched-chain acyl-coA, were significantly downregulated, so were multiple enzymes involved in downstream reactions for converting the CoA products into the TCA cycle (Fig. 1b, c). The mRNA level of mitochondrial targeted 2C-type serine/threonine protein phosphatase (PP2Cm), that dephosphorylates and activates BCKDH complex, was also decreased (Fig. 1b, c). The protein levels of the BCAT2, branched-chain keto acid dehydrogenase E1 Alpha Polypeptide (BCKDHA), and PP2Cm were also significantly decreased in Glut1-TG hearts (Fig. 1d). No reduction was seen in the mRNA and protein expression of BCKD kinase (BCKDK) in Glut1-TG hearts (Supplementary Fig. 1c, d).

Consistent with the in vivo observations in mice, increasing glucose uptake by either overexpression of Glut1 or high glucose medium (HG, 25 mM) reduced the expressions of BCAT2, BCKDHA, and PP2Cm, the three key proteins in BCAA degradation pathway, in both primary cells and proliferating cell lines (Fig. 1e, Supplementary Fig. 1e-h). Taken together, these data indicate that high glucose suppresses the expression of BCAA degradation enzymes.

We subsequently performed targeted metabolomics analysis to determine whether the downregulation of gene expression in Glut1-TG hearts affected the BCAA degradation in vivo. Increased glucose uptake in Glut1-TG hearts led to a higher level of glycolytic metabolites (Supplementary Fig. 1i, j), as expected. Importantly, the level of BCAAs was also increased while the downstream BCAA metabolites, e.g., α-keto-β-methylvalerate (KMV) and α-ketoisocaproate (KIC) were decreased in Glut1-TG hearts (Fig. 1f). Notably, levels of other amino acids exhibited the opposing trend compared to that of the BCAAs (Supplementary Fig. 1i), suggesting that high glucose selectively inhibited BCAA degradation. Independent biochemical assays showed a 2.5-fold increase of BCAA level in Glut1-TG hearts (Fig. 1g). In addition, high glucose medium or Glut1 overexpression also increased the BCAA content in both myocyte and non-myocyte cell types (Fig. 1h, Supplementary Fig. 1k). Taken together, these observations strongly suggest that glucose negatively regulates intracellular BCAA degradation.

**KLF15 is the target of high intracellular glucose.** We next searched for the upstream regulatory mechanisms mediating the transcriptional suppression of BCAA degradation pathway. The microarray data showed ~2-fold downregulation of *KLF15* in Glut1-TG hearts (Supplementary Fig. 2a), which was confirmed by RT-PCR and immunoblot (Fig. 2a, b). Furthermore, both high glucose medium and Glut1 overexpression reduced the *KLF15* mRNA and protein levels in neonatal rat cardiomyocytes (NRCMs) (Fig. 2c, d, Supplementary Fig. 2b). Similar observations were also made in other cell lines including H9C2 and HEK-293 (Supplementary Fig. 2c, d). Previously, KLF15 has been shown to mediate the transcription of genes for BCAA degradation in multiple organs, including skeletal muscle, liver, and heart[27,28]. We found that overexpression of KLF15 in the presence of high glucose is sufficient to normalize both mRNA and protein expression of BCAA degradation enzymes (Fig. 2e, Supplementary Fig. 2e, f), suggesting that glucose negatively regulates BCAAs degradation through a KLF15 dependent mechanism.

To investigate the mechanisms by which high glucose suppressed KLF15 expression, the *KLF15* promoter region containing ~1 kb upstream sequence from the transcription start site or its serially truncated segments was cloned into a luciferase vector and the luciferase activity was examined (Fig. 3a). High glucose medium or Glut1 overexpression significantly reduced

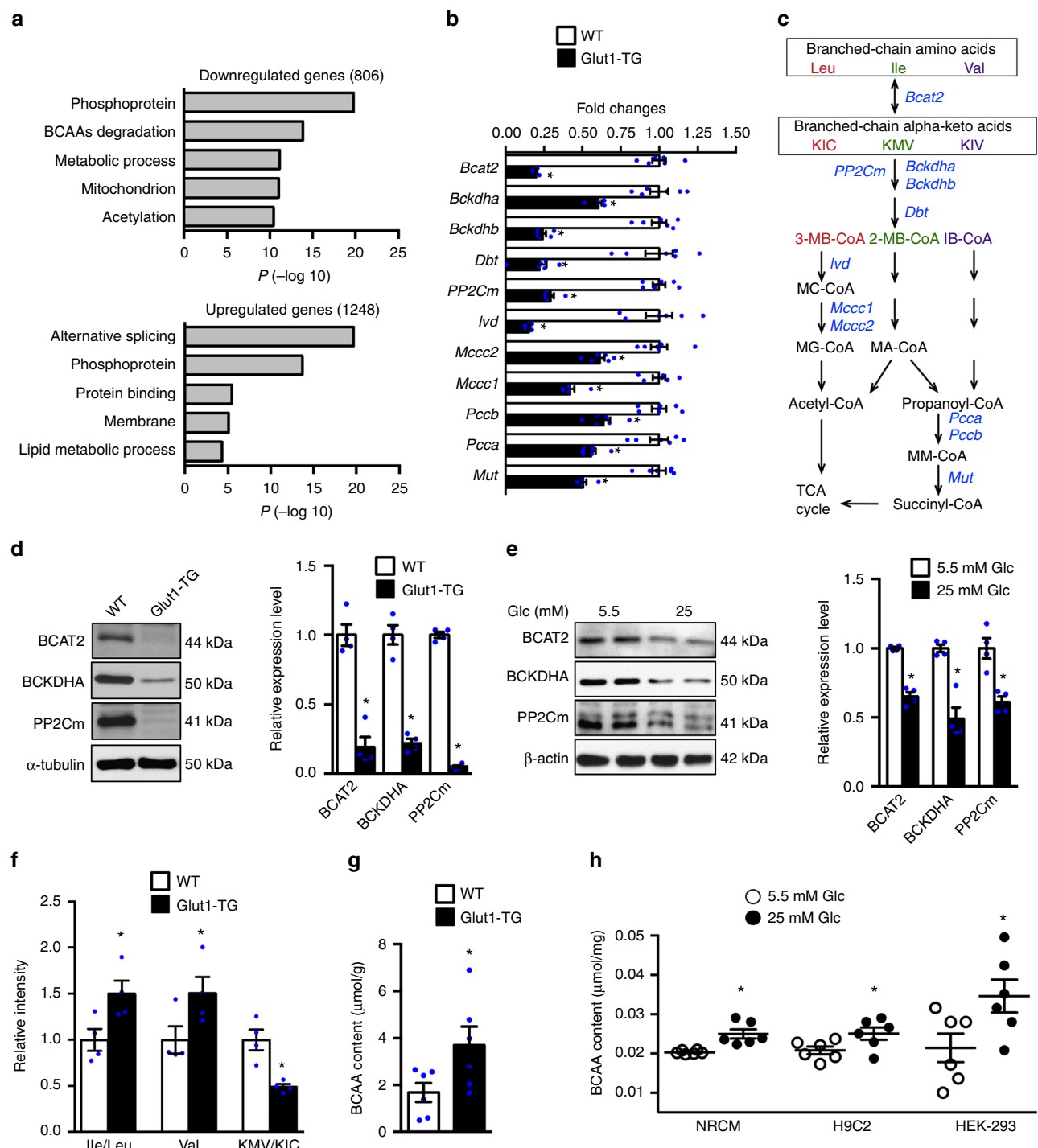

the luciferase activity containing 1 kb promoter sequence and the combination of the two showed synergetic effects (Supplementary Fig. 2g). However, the suppression effect on luciferase activity by high glucose was no longer observed in smaller promoter constructs (Fig. 3a), indicating that the glucose response element(s) is located between the −1068 to −541 bp regions. We found two potential binding elements (CREI and CREII) for the cAMP response element binding protein (CREB) in this region (http://jaspar.genereg.net/); both exhibited high similarity with CRE consensus sequence and were conserved across species (Fig. 3b). The chromatin immunoprecipitation (ChIP) revealed that endogenous CREB bound to both CREI and CREII sites

(Fig. 3b). Importantly, high glucose significantly reduced CREB binding occupancy of CREI and CREII on the *KLF15* promoter (Fig. 3c). To further test if high glucose suppressed the CREB transcriptional activity, cells expressing a luciferase vector containing three CREB consensus elements (CRE-luc) were subjected to high glucose medium or Glut1 overexpression. Both measures significantly decreased CRE luciferase activity (Fig. 3d). We also observed that the phosphorylation of CREB at Ser 133, critical for CREB transcriptional activation[29], was significantly decreased in cells cultured in high glucose medium or over-expressing Glut1 (Fig. 3e). Moreover, overexpression of CREB was sufficient to attenuate high glucose induced downregulation

**Fig. 1** Glucose negatively regulates BCAA degradation. **a** GO enrichment analysis of differentially regulated genes in Glut1-TG hearts using DAVID. Top 5 clusters of either downregulated or upregulated genes are shown. **b** qRT-PCR analyses of selected BCAA degradation genes in Glut1-TG and WT mouse hearts. The expression was normalized to 18S rRNA and reported as fold change over WT (*$p < 0.05$ vs. WT, $n = 6$). Bckdhb branched-chain keto acid dehydrogenase E1 subunit beta, Dbt dihydrolipoamide branched-chain transacylase E2, Ivd isovaleryl-CoA dehydrogenase, Mccc2 Methylcrotonoyl-CoA Carboxylase 2, Mccc1 Methylcrotonoyl-CoA Carboxylase 1, Pccb Propionyl Coenzyme A Carboxylase, Beta Polypeptide; Pcca Propionyl Coenzyme A Carboxylase, Alpha Polypeptide; Mut Methylmalonyl Coenzyme A Mutase. **c** Schematic illustration of the BCAA degradation pathway. Enzymes examined in Fig. 1b are shown in blue. Degradation of Leu, Ile, and Val share the same initial steps catalyzed by Bcat2, Bckdha, Bckdhb, PP2Cm, and Dbt. Leu leucine, Ile isoleucine, Val valine, KIV α-ketoisovalerate, 3-MB-CoA 3-Methylbutanoyl-CoA, 2-MB-CoA 2-Methylbutanoyl-CoA, IB-CoA Isobutyryl-CoA, MC-CoA 2-Methylcrotonoyl-CoA, MG-CoA 2-Methylglutaconyl-CoA, MA-CoA 2-Methylbutanoyl-CoA, MM-CoA Methylmalonyl-CoA. **d** Representative immunoblots of BCAT2, BCKDHA, PP2Cm, and α-tubulin in heart tissue homogenates (left) and statistical analyses of densitomeric measurements of BCAT2, BCKDHA, and PP2Cm (right) are shown (*$p < 0.05$ vs. WT, $n = 4$). **e** NRCMs were incubated with DMEM containing 5.5 mM or 25 mM glucose for 24 h. Representative immunoblots of cell lysates (left) and statistical analyses of densitomeric measurements of BCAT2, BCKDHA, and PP2Cm (right) are shown (*$p < 0.05$ vs. 5.5 mM Glc, $n = 4$). Glc glucose. **f** The relative intensity of indicated BCAAs and their metabolites measured by targeted metabolomics of Glut1-TG and WT mouse hearts (*$p < 0.05$ vs. WT, $n = 4$). **g** The intracellular BCAA concentration was quantified in Glut1-TG and WT hearts (*$p < 0.05$ vs. WT, $n = 6$). **h** Cellular BCAA levels after 24 h incubation with DMEM containing 5.5 mM or 25 mM glucose (*$p < 0.05$ vs. 5.5 mM Glc, $n = 6$). Data shown as mean ± s.e.m. P values were determined using unpaired Student's t-test (**b, d, e, f, g, h**) or Mann–Whitney test (**b, d, e, f**)

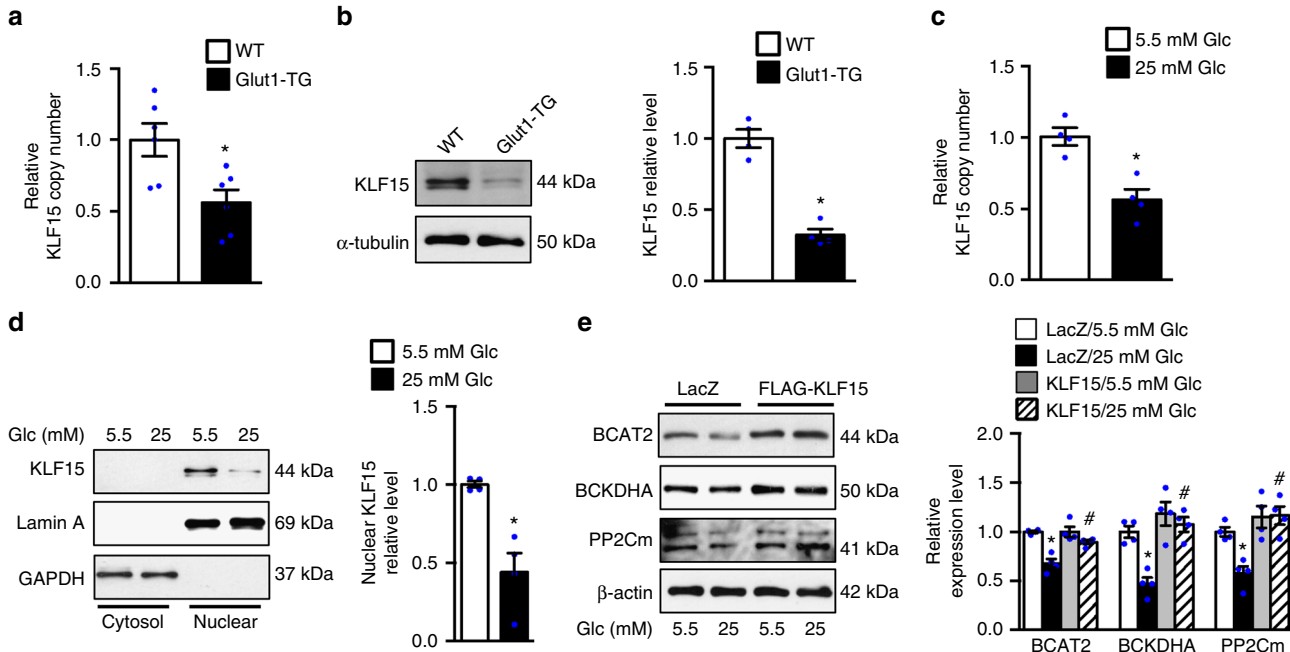

**Fig. 2** KLF15 is essential for glucose-mediated downregulation of BCAA degradation. **a** qRT-PCR analysis of KLF15 mRNA in Glut1-TG and WT hearts. The expression was normalized to 18S rRNA and reported as fold change over WT (*$p < 0.05$ vs. WT, $n = 6$). **b** Representative immunoblots of KLF15 and α-tubulin in heart tissue homogenates (left) and statistical analysis of densitomeric measurement of KLF15 (right) are shown (*$p < 0.05$ vs. WT, $n = 4$). **c** The mRNA level of KLF15 in NRCMs incubated with DMEM containing 5.5 mM or 25 mM glucose for 24 h (*$p < 0.05$ vs. 5.5 mM Glc, $n = 4$). **d** NRCMs were incubated with DMEM containing 5.5 mM or 25 mM glucose for 24 h. Representative immunoblots from cytosolic and nuclear fractions (left) and statistical analysis of densitomeric measurement of nuclear KLF15 (right) are shown (*$p < 0.05$ vs. 5.5 mM Glc, $n = 4$). **e** NRCMs transduced with indicated adenovirus were incubated with DMEM containing 5.5 mM or 25 mM glucose for 24 h. Immunoblots of cell lysates (left) and statistical analyses of densitomeric measurements of BCAT2, BCKDHA, and PP2Cm (right) are shown (*$p < 0.05$ vs. LacZ/5.5 mM Glc, #$p < 0.05$ vs. LacZ/25 mM Glc, $n = 4$). Data shown as mean ± s.e.m. P values were determined using unpaired Student's t-test (**a, b, c, d**) or one-way ANOVA followed by Newman–Keuls comparison test (**e**)

of KLF15, as well as BCAA degradation enzymes (Fig. 3f, g). These data indicated that high glucose inhibited *KLF15* transcription by negatively regulating the binding of CREB to its promoter region.

**Glucose promotes cell growth through the KLF15-BCAA pathway.** Previous studies have identified KLF15 as a negative regulator of cardiac hypertrophy[21]. In pathological cardiac hypertrophy caused by pressure overload in vivo or by phenylephrine (PE) treatment of NRCMs in vitro, the expressions of

KLF15 and BCAA degradation enzymes were decreased (Supplementary Fig. 3a–d), whereas glucose utilization increased[6,30]. Physiological stimuli for growth that caused increased glucose uptake e.g., insulin or insulin like growth factor 1 (IGF-1) also induced similar changes of KLF15 and BCAA degradation enzymes (Supplementary Fig. 3e–g)[31]. While overexpression of Glut1 did not further increase myocyte cell size in response to PE stimulation (Supplementary Fig. 3h), reducing glucose uptake by knocking down Glut1 (sh-Glut1, Supplementary Fig. 3i, j) was sufficient to attenuate the downregulation of KLF15 and BCAA

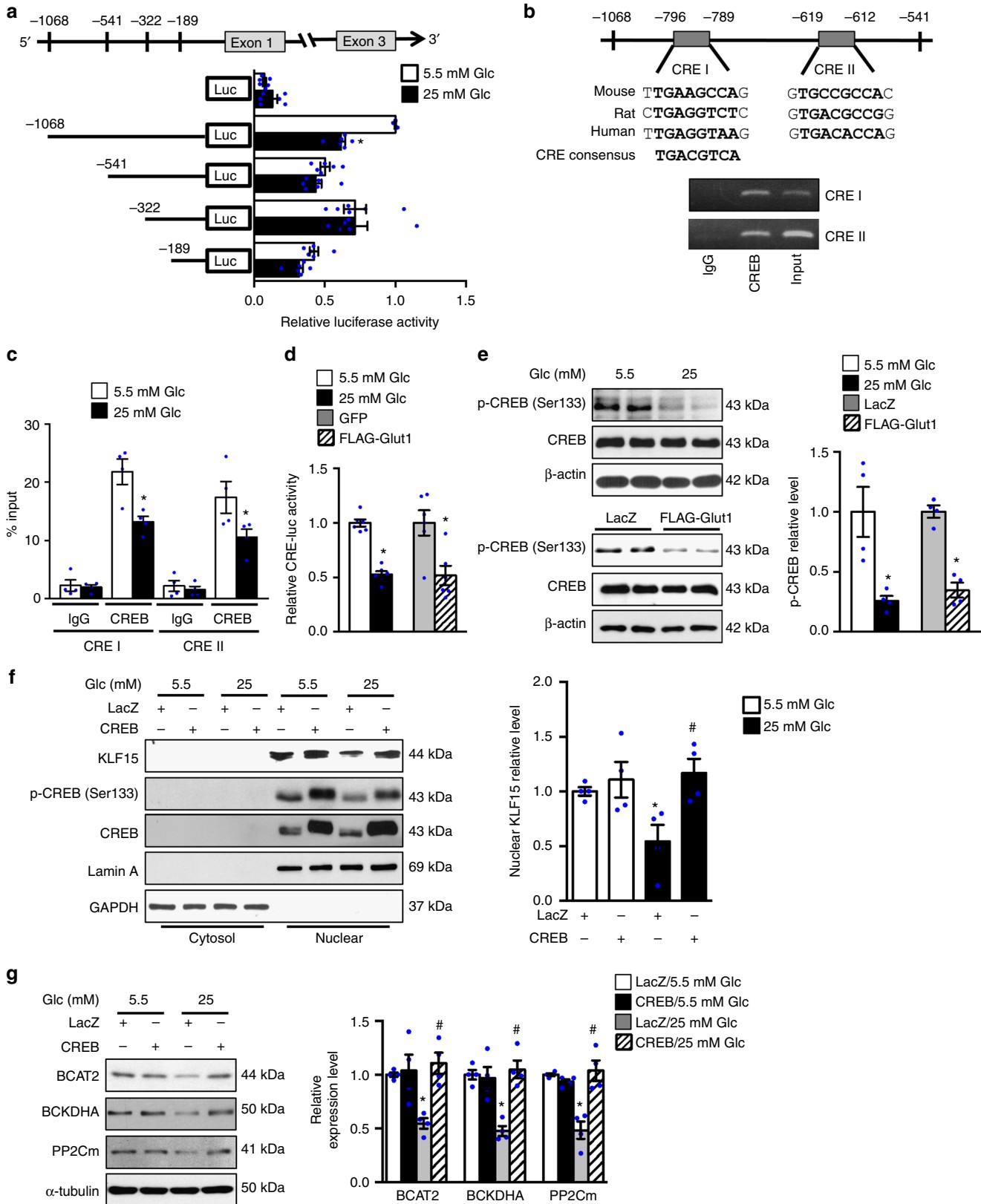

degradation enzymes in PE stimulated cells (Fig. 4a, b). Furthermore, the increases in cell size and atrial natriuretic factor (ANF) promoter activity was significantly reduced by knocking down Glut1 (sh-Glut1) in the presence of PE (Fig. 4c, d). Knockdown of KLF15 reversed the anti-hypertrophy effect by

sh-Glut1 in PE treated groups (Fig. 4c, d, Supplementary Fig. 3k), suggesting that KLF15 was an important mediator of glucose-dependent cell growth. We also found that overexpression of KLF15 prevented the downregulation of BCAA degradation enzymes in PE treated cells, in parallel with the reductions in cell

**Fig. 3** Glucose negatively regulates KLF15 through a CREB dependent mechanism. **a** NRCMs transfected with 1 μg indicated KLF15 promoter luciferase vectors or empty vector were incubated with DMEM containing 5.5 mM or 25 mM glucose for 36 h. The luciferase activity was measured (*$p < 0.05$ vs. −1068-luc/5.5 mM Glc, $n = 6$). **b** Upper panel: a schematic illustration of two potential CREB DNA-binding elements in the KLF15 promoter region, referred to as CRE I and CRE II. Lower panel: ChIP analysis of CREB binding of the KLF15 promoter in vivo. Results are representative of three independent experiments. **c** NRCMs were incubated with DMEM containing 5.5 mM or 25 mM glucose for 24 h. Protein-bound chromatin was prepared and immunoprecipitated with IgG and CREB antibodies. The relative occupancy on the promoter was compared with the input signal (*$p < 0.05$ vs. 5.5 mM Glc, $n = 4$). **d** NRCMs transfected with 1 μg CRE-luc vector were incubated with DMEM containing 5.5 mM or 25 mM glucose or transfected with Glut1 or control (GFP) plasmid for 36 h. The luciferase activity was measured (*$p < 0.05$ vs. 5.5 mM Glc or GFP, $n = 6$). **e** NRCMs were transduced with indicated adenovirus or incubated with DMEM containing 5.5 mM or 25 mM glucose for 24 h. Representative immunoblots of cell lysates for p-CREB (Ser 133), CREB and β-actin (left) and statistical analysis of densitomeric measurement of p-CREB (Ser 133) (right) are shown (*$p < 0.05$ vs. 5.5 mM Glc or LacZ, $n = 4$). **f**, **g** NRCMs transduced with indicated adenovirus were incubated with DMEM containing 5.5 mM or 25 mM glucose for 24 h. **f** Representative immunoblots of cytosolic and nuclear fractions (left) and statistical analysis of densitomeric measurement of nuclear KLF15 (right) are shown (*$p < 0.05$ vs. LacZ/5.5 mM Glc, #$p < 0.05$ vs. LacZ/25 mM Glc, $n = 4$). **g** Immunoblots of cell lysates (left) and statistical analyses of densitomeric measurements of BCAT2, BCKDHA and PP2Cm (right) are shown (*$p < 0.05$ vs. LacZ/5.5 mM Glc, #$p < 0.05$ vs. LacZ/25 mM Glc, $n = 4$). Data shown as mean ± s.e.m. *P* values were determined using unpaired Student's *t*-test (**a**, **d**, **e**), Mann–Whitney test (**c**, **d**), or one-way ANOVA followed by Newman-Keuls comparison test (**f**, **g**)

size and ANF promoter luciferase activity (Fig. 4e–g, Supplementary Fig. 3l). The anti-hypertrophy effect of KLF15 overexpression was partially abolished when BCAA degradation pathway was inhibited by knocking down PP2Cm or BCAT2 (Fig. 4f, g, Supplementary Fig. 3m, n). Taken together, these data reveal a mechanism that links glucose reliance in cell growth to BCAA degradation pathway through KLF15.

**Glucose modulates mTOR in an energy independent manner.** In searching for the end effector of glucose mediated cell growth, we found that knockdown of Glut1 attenuated PE induced mTOR activation in NRCMs (Supplementary Fig. 4a). Consistent with previous observations, inhibition of mTOR activity by rapamycin treatment is sufficient to prevent PE induced cardiomyocyte growth (Supplementary Fig. 4b, c). As glucose is an important energy substrate and the mTOR activity is highly sensitive to cellular energy status, we investigated whether glucose is required to maintain cellular energy balance during the growth. Replacing glucose with either pyruvate (12 mM) or lactate (12 mM) in the culture medium attenuated NRCMs' growth in response to hypertrophic stimulus (Fig. 5a) with no change in viability or intracellular ATP level compared with cells cultured with glucose (Supplementary Fig. 4d, e). Moreover, only NRCMs cultured with glucose showed mTOR activation in response to PE treatment (Fig. 5b, Supplementary Fig. 4f). However, we observed that NRCMs cultured with pyruvate or lactate exhibited increased phosphorylation of AMP-activated protein kinase (AMPK) indicating that AMPK was activated (Supplementary Fig. 4f). AMPK is an energy sensor and a negative regulator of mTOR activity and cell growth[32]. To determine whether restricting glucose inhibited mTOR through activation of AMPK, we prevented AMPK activation by including both lactate (6 mM) and acetate (6 mM) as the replacement of glucose in the culture (Supplementary Fig. 4d, e, Fig. 5b). NRCMs cultured with lactate and acetate, nevertheless, remained resistant to mTOR activation and hypertrophic growth by PE (Fig. 5a, b), suggesting that AMPK activation is not required for mTOR inhibition under these conditions. Similarly, rat adult cardiomyocytes cultured with pyruvate or lactate showed no activation of AMPK but showed attenuated mTOR activation and reduced hypertrophic growth during PE stimulation (Supplementary Fig. 4g, h). Taken together, these observations show that independent of its function in energy provision, glucose plays an essential role in mediating mTOR activation in response to growth stimulation.

**mTOR activation requires the inhibition of BCAA degradation.** To test if the suppression of BCAA degradation by glucose was required for mTOR activation during cell growth, we first

examined the intracellular BCAAs amount in response to PE treatment at various time points. PE induced a transient accumulation of intracellular BCAA that coincided with activation of mTOR, which was prevented by overexpression of KLF15 (Fig. 6a, b, Supplementary Fig. 4i). This transient accumulation of BCAA in cultured cardiomyocytes is consistent with the observation in the hearts in response to pressure overload or myocardial infarction, in which the increase of myocardial BCAA level peaks at one week after the stimulus[33,34]. Knockdown of BCAT2 or PP2Cm, the key enzymes for BCAA degradation, induced the intracellular BCAA accumulation (Supplementary Fig. 4j) and restored mTOR activation by PE in KLF15 overexpressing cells (Fig. 6c, Supplementary Fig. 4k). Similarly, the downregulation of BCAA degradation enzymes by PE was abolished in cardiomyocytes cultured with non-glucose medium (Supplementary Fig. 4l). Knocking down PP2Cm or BCAT2 was sufficient to reactivate mTOR during PE treatment in both neonatal and adult rat cardiomyocytes cultured even in non-glucose medium and restored the growth response to the same extent as in cells cultured with glucose (Fig. 6d, e, Supplementary Fig. 4m-p). Taking together, these data suggest that suppression of BCAA degradation pathway by glucose is required for mTOR activation during cell growth.

**Crosstalk between growth stimulus and metabolic response.** Although glucose is required for cell growth, high glucose alone did not stimulate mTOR activation in cardiomyocytes in the absence of hypertrophic stimuli (Supplementary Fig. 5a). The Glut1-TG hearts did not develop hypertrophy without stress[17], and high glucose medium did not increase cell size in NRCMs (Supplementary Fig. 5b). These findings suggest that BCAA accumulation induced by increased glucose reliance is required but not sufficient to stimulate cell growth, i.e., the activation of mTOR requires additional input even in the presence of metabolic reprogramming. We thus investigated the cross talk between glucose and the two upstream regulators of mTOR signaling: PI3K and MAPK pathways. As expected, PE treatment stimulated both PI3K and MAPK pathways (Supplementary Fig. 5c), while high glucose had no effect on either (Supplementary Fig. 5d). Similarly, removing glucose or overexpression of KLF15 had no effect on activation of either PI3K or MAPK in response to PE treatment (Fig. 7a, b, Supplementary Fig. 5e), but strongly suppressed mTOR activation and cardiomyocyte growth in response to PE (Figs. 4f, 5a, b, 6c, Supplementary Fig. 4i). In addition, inhibition of PI3K pathway by wortmannin or inhibition of MAPK by U0126 was sufficient to inhibit mTOR activation during PE stimulation but had no effect on BCAA degradation pathway (Fig. 8a, Supplementary Fig. 5f).

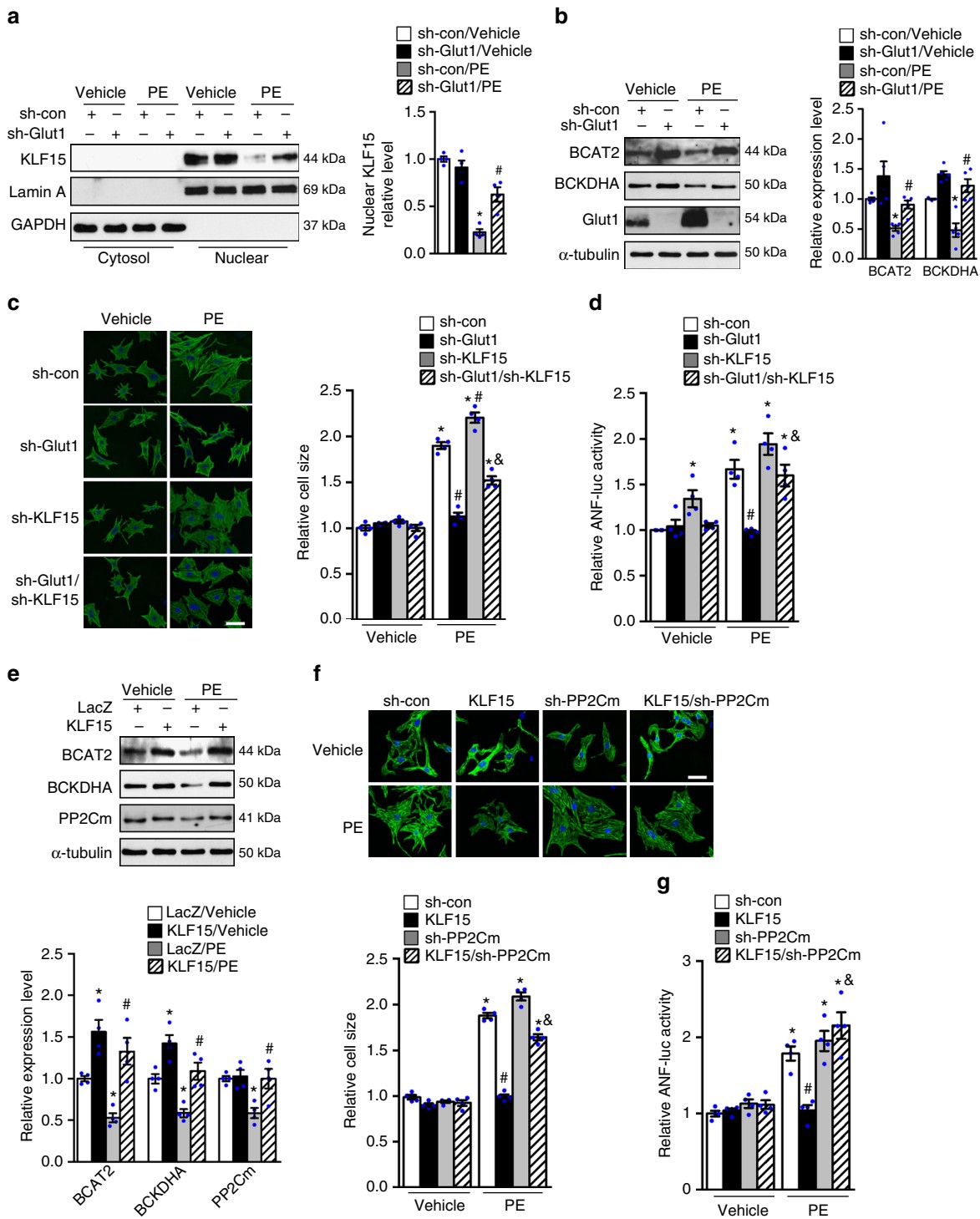

Knockdown of KLF15 or PP2Cm in cells subjected to PE stimulation, which effectively suppressed BCAA degradation pathway, failed to restore mTOR activation or hypertrophic growth when PI3K or MAPK was inhibited (Fig. 8b, c, Supplementary Fig. 5g–l), indicating that PE induced mTOR activation also required intact PI3K or MAPK signaling. Taken together, these data suggest that the activation of mTOR during a growth response requires input signal via the PI3K and/or MAPK pathway, as well as the metabolic response via increased intracellular glucose (Fig. 8d). Either component is indispensable, and they act cooperatively to sustain a growth response.

**Glucose-BCAA circuit regulates cardiac hypertrophy in mice.** Increased glucose reliance, downregulation of KLF15 and impaired BCAA degradation have been demonstrated in hearts with pathological hypertrophy[6,21,28], but their role as a regulatory circuit for hypertrophic response has never been tested. If our hypothesis is correct, the anti-hypertrophy effect of KLF15 will not be observed in mice deficient of BCAA degradation such as PP2Cm KO. We thus sought to increase KLF15 expression in the heart of WT and PP2Cm KO mice, via retro-orbital injection of an adeno-associated virus serotype 9 vector carrying KLF15 and directed by the cardiac-specific cTNT promoter (AAV9-KLF15)[35].

**Fig. 4** Suppression of BCAA degradation by high glucose is required for cardiomyocyte growth in response to PE. **a**, **b** NRCMs transduced with indicated adenovirus for 72 h were treated with phenylephrine (PE, 100 μM) or vehicle for 6 h. **a** Representative immunoblots of cytosolic and nuclear fractions (left) and statistical analysis of densitomeric measurement of KLF15 (right) are shown (*$p < 0.05$ vs. sh-con/Vehicle, #$p < 0.05$ vs. sh-con/PE, $n = 4$). **b** Representative immunoblots of cell lysates (left) and statistical analyses of densitomeric measurements of BCAT2 and BCKDHA (right) are shown (*$p < 0.05$ vs. sh-con/Vehicle, #$p < 0.05$ vs. sh-con/PE, $n = 5$). **c** NRCMs transduced with indicated adenovirus were treated with phenylephrine (PE, 100 μM) or vehicle for 48 h. Cellular surface area in each group was quantified and expressed relative to the control (*$p < 0.05$ vs. sh-con/Vehicle, #$p < 0.05$ vs. sh-con/PE, &$p < 0.05$ vs. sh-Glut1/PE, $n = 4$). Scale bar, 25 μm. **d** NRCMs transfected with 1 μg ANF promoter luciferase reporter (ANF-luc) were transduced with indicated adenovirus for 36 h and further incubated with phenylephrine (PE, 100 μM) or vehicle. The luciferase activity was measured after 36 h of PE (*$p < 0.05$ vs. sh-con/Vehicle, #$p < 0.05$ vs. sh-con/PE, &$p < 0.05$ vs. sh-Glut1/PE, $n = 4$). **e** NRCMs transduced with indicated adenovirus were treated with phenylephrine (PE, 100 μM) or vehicle for 6 h. Representative immunoblots of cell lysates (upper) and statistical analyses of densitomeric measurements of BCAT2, BCKDHA and PP2Cm (lower) are shown (*$p < 0.05$ vs. LacZ/Vehicle, #$p < 0.05$ vs. LacZ/PE, $n = 4$). **f** NRCMs transduced with indicated adenovirus were treated with phenylephrine (PE, 100 μM) or vehicle for 48 h. Cellular surface area in each group was quantified and expressed relative to the control (*$p < 0.05$ vs. sh-con/Vehicle, #$p < 0.05$ vs. sh-con/PE, &$p < 0.05$ vs. KLF15/PE, $n = 4$). Scale bar, 25 μm. **g** NRCMs transfected with 1 μg ANF-luc were transduced with indicated adenovirus followed by incubation with phenylephrine (PE, 100 μM) or vehicle for 36 h. The luciferase activity was measured (*$p < 0.05$ vs. sh-con/Vehicle, #$p < 0.05$ vs. sh-con/PE, &$p < 0.05$ vs. KLF15/PE, $n = 4$). Data shown as mean ± s.e.m. $P$ values were determined using one-way ANOVA followed by Newman–Keuls comparison test (**a**, **b**, **g**) or Kruskal–Wallis test followed by Dunn's comparison test (**c**, **d**, **e**, **f**)

This approach showed high efficiency and specificity of gene expression in the heart (Supplementary Fig. 6a, b). As high level overexpression of KLF15 in transgenic mice causes arrhythmia due to a regulatory role of KLF15 on potassium channel[36], we titrated the dosage of AAV9-KLF15 to limit the KLF15 over-expression to ~2-fold (Fig. 9a, Supplementary Fig. 6c). The expression level of KLF15 was stable for up to 5 weeks and did not cause any overt phenotype in WT mice. Importantly, AAV9-KLF15 administration was sufficient to increase multiple enzymes involved in BCAA degradation pathway (Fig. 9a).

We found that the increases in heart weight and myocyte cross-sectional area, induced by TAC, were significantly reduced by KLF15 overexpression in WT but not in *PP2Cm* KO mice (Fig. 9b, c). Immunoblot analysis demonstrated that AAV9-KLF15 treatment significantly reduced the phosphorylation of p70 S6K and mTOR after TAC in WT mice but not in *PP2Cm* KO mice (Fig. 9d). Importantly, PI3K and MAPK signaling was activated to a similar extent in all the groups subjected to TAC compared with sham operated mice (Supplementary Fig. 6c). In addition, molecular markers for pathological cardiac hypertrophy *ANP* and *BNP*, were markedly elevated in WT-TAC and a greater increase observed in *PP2Cm* KO-TAC hearts (Fig. 9e). AAV9-KLF15 treatment significantly reduced the expression of *ANP* and *BNP* in WT but not in *PP2Cm* KO hearts after TAC (Fig. 9e). Echocardiographic measurements demonstrated that TAC induced cardiac dysfunction in both WT and *PP2Cm* KO mice compared with sham operated mice, which was rescued by AAV9-KLF15 in WT only (Fig. 9f, Supplementary Table 2). Collectively, these data demonstrate that suppression of KLF15-BCAA degradation pathway is required for activation of mTOR and development of pathological hypertrophy of adult hearts in vivo.

## Discussion

The study reveals a regulatory circuit between glucose and BCAA that serves as a metabolic determinant of growth signaling. In a variety of cell types, we observed that increased intracellular glucose inhibits the expression of KLF15 and its downstream genes resulting in decreased BCAA degradation. Such a response is required for sustaining the activation of mTOR during growth stimulation even though energy provision by glucose is dispensable. Importantly, disruption of this regulatory cascade alleviates cardiomyocyte hypertrophy during pressure overload in mice demonstrating that this mechanism plays a critical role in cellular growth response in disease.

The mTOR complex is a nutrient sensor and a signaling hub where various inputs are vetted and relayed cumulating in changes of protein synthesis, cell growth, and survival. Here we demonstrate that activation of mTOR requires growth stimuli that act in concert with increased glucose metabolism. This is consistent with previous observations that mTOR activation by insulin stimulation or increased hemodynamic load in the heart requires increased glucose metabolism[15,16]. However, prior hypothesis that mTOR activity is modulated by a direct interaction of glucose metabolite(s) with protein(s) in the mTOR complex has not been supported by direct evidence. Here we have identified a cascade of mechanisms linking glucose metabolism to the mTOR activity through the inhibition of BCAA degradation. Previous studies have shown that mTOR activity is regulated by nutrients, mostly in the form of amino acids, especially leucine, the most abundant BCAA[37–40]. Here we demonstrate that increased glucose metabolism in growing cells modulates the amino acid levels by suppressing the BCAA degradation pathway, thus, reveal a missing link in the coordination of cell metabolism and intracellular nutrient environment for cell growth. Furthermore, the identification of the glucose-BCAA circuit reveals a fundamental mechanism in metabolic regulation that will likely advance research on metabolic signaling beyond cell growth.

Similar to our observations in this study, activation of mTOR by amino acids also requires the presence of upstream signals, such as growth factors[39]. These observations indicate that metabolic responses elicited by growth signals, instead of independently stimulating cell growth, play an obligatory role in setting the intracellular environment in favor of growth signal transduction. Understanding the molecular mechanisms underlying the metabolic determinant of mTOR activity, therefore, presents new avenue to target cell growth independent of growth stimulation.

We find that restoration of mTOR activity is sufficient to promote the growth of both neonatal and adult rat cardiomyocytes in the absence of glucose. This seemingly surprising observation suggests that glycolysis is not required in the growth of post-mitotic cells once mTOR is activated. Substitution of glucose with pyruvate, lactate and/or, acetate is sufficient to meet the energy demand and prevent AMPK activation. However, glucose metabolites also fuel several accessary pathways, e.g., hexosamine biosynthetic pathway or pentose phosphate pathway. Since gluconeogenesis is normally not present in cardiomyocytes, long term culture with non-glucose substrates could deplete metabolites produced by these pathways which have been shown critical for cell growth[3,41]. It is possible that these metabolites are

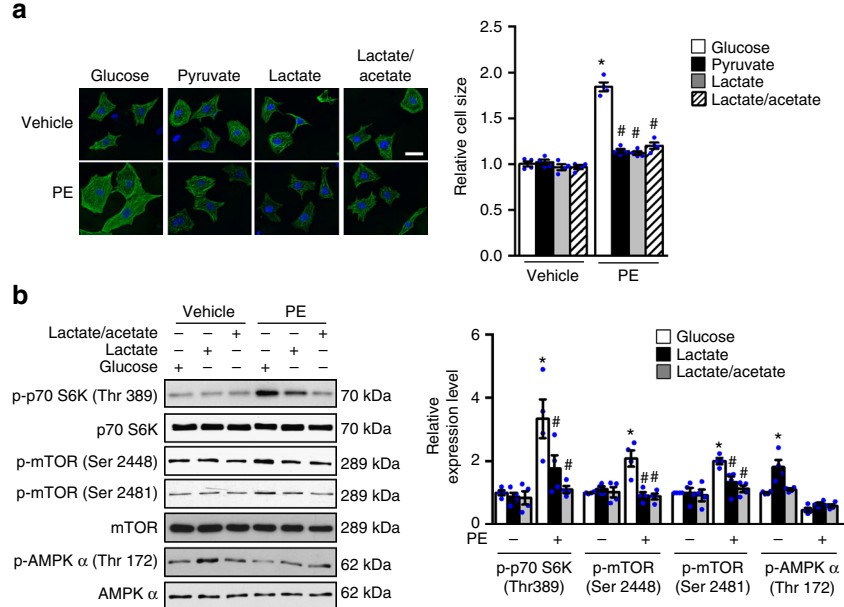

**Fig. 5** Glucose removal inhibits mTOR activation and cell growth. **a** NRCMs were pretreated with DMEM containing either glucose or non-glucose substrates for one hour and then incubated with phenylephrine (PE, 100 μM) or vehicle for 48 h. Myocytes were fixed and stained with anti-Troponin T to visualize CMs. Cellular surface area in each group was quantified and expressed relative to the control (*$p < 0.05$ vs. Glucose/Vehicle, #$p < 0.05$ vs. Glucose/PE, $n = 4$). Scale bar, 25 μm. **b** NRCMs were pretreated with DMEM containing either glucose or non-glucose substrates for one hour and then incubated with phenylephrine (PE, 100 μM) or vehicle for 6 h. Immunoblots of cell lysates (left) and statistical analyses of densitomeric measurements of p-p70 S6K (Thr 389), p-mTOR (Ser 2448), p-mTOR (Ser 2481), and p-AMPK α (Thr 172) (right) are shown (*$p < 0.05$ vs. Glucose/PE(-), #$p < 0.05$ vs. Glucose/PE(+), $n = 4$). Data shown as mean ± s.e.m. $P$ values were determined using one-way ANOVA followed by Newman–Keuls comparison test (**b**) or Kruskal-Wallis test followed by Dunn's comparison test (**a**)

not sufficiently depleted during the short-term experiments in the present study hence not affecting cell growth. It also raises the intriguing question whether other mechanisms are present to compensate for the loss of glycolysis in those cells during hypertrophic growth. Further investigations along this line may lead to discovery of novel metabolites and pathways that contribute to the growth of post-mitotic cells.

A key finding of this study is the identification of KLF15 as a critical node connecting metabolism and cell growth. KLF15 has long been recognized as a pleiotropic transcriptional factor[20,42]. KLF15-mediated transcriptional mechanisms regulate cell differentiation, circadian rhythm, ion channel activity, and multiple aspects of metabolism[23,27,36,43–45]. Despite clear evidence for a role of KLF15 in determining the metabolic phenotype of differentiated cells, the upstream mechanism(s) that drive KLF15 expression have not been defined. Interestingly, KLF15 expression is substantially suppressed during development and in proliferating cells when basal glucose uptake and utilization is high[21–23,46]. Results from this study reveal a regulatory mechanism by glucose through which immature cells suppress KLF15 to sustain a metabolic profile favorable for cell growth. Consistent with the notion, exponential increases of KLF15 expression occur at the end of postnatal growth period which coincide with the switch to insulin regulated glucose disposal in the heart coupled with increased fatty acid oxidation and amino acid catabolism[23,24]. When the metabolic pattern returned to the fetal form in hypertrophic hearts, KLF15 level is reduced[21]. Indeed, the present study confirmed prior observation that both physiological and pathological growth stimuli, known to stimulate glucose uptake, reduced the KLF15 expression[21,47]. Here we have identified one glucose responsive mechanism that suppresses KLF15 transcription through reducing the binding of CREB on its promoter region. KLF15 expression is also responsive to multiple hormonal regulations such as glucocorticoids or

during fasting of which the molecular mechanisms are unknown. The role of intracellular glucose in these responses warrants investigation.

Several observations have been made that the expression of KLF15 and BCAA degradation enzymes are downregulated in failing hearts[21,28]. However, Glut1-TG mice was able to maintain cardiac function at baseline and protected against pressure overload induced cardiac dysfunction even though the KLF15-BCAA pathway was inhibited. We speculate that the seemingly contradictory outcome observed in Glut1-TG hearts reflects the complex role of substrate metabolism in modulating myocardial energetics and signaling transduction. In Glut1-TG mice, the tremendous capacity of glucose supply allows glucose to become the predominant fuel and is able to maintain cardiac energetics during pressure overload, a condition when downregulation of fatty acid oxidation (FAO) compromises myocardial energy supply while increased glucose utilization through endogenous mechanisms is not sufficient to maintain myocardial energetics[17,48]. The study by Sun et al. suggests that accumulation of BCAA metabolites, BCKAs, contribute to the development of TAC induced cardiac dysfunction in *PP2Cm* KO hearts[28]. We did not observe BCKAs accumulation in Glut1-TG hearts (Fig. 1f), since the entire BCAA degradation pathway was downregulated including the enzyme converting BCAAs to BCKAs. Therefore, improved energy metabolism in Glut1-TG hearts likely contributes to delayed progression to heart failure. However, compared to other interventions of sustaining myocardial energetics, such as preventing the downregulation of FAO, improved function in Glut1-TG after TAC was not accompanied by the expected reduction of cardiac hypertrophy[49]. These observations are consistent with the role of glucose in regulating cell growth independent of its role in ATP production, as shown in the present study.

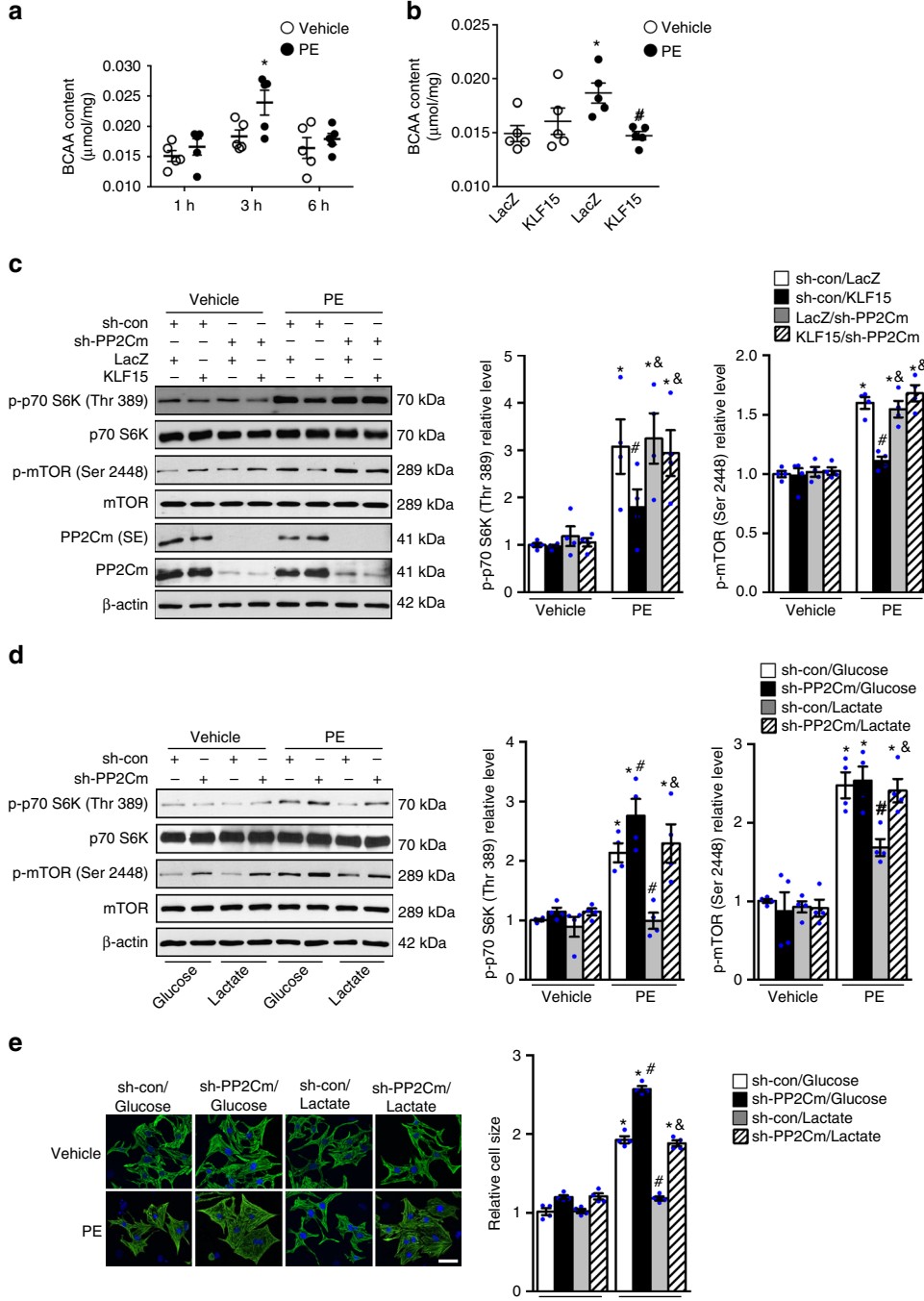

**Fig. 6** BCAA accumulation is required for mTOR activation during cardiomyocyte growth. **a** Cellular BCAA level was measured in NRCMs treated with phenylephrine (PE, 100 μM) or vehicle at indicated time points (*$p < 0.05$ vs. Vehicle, $n = 5$). **b** Cellular BCAA level was examined in NRCMs transduced with indicated adenovirus and treated with phenylephrine (PE, 100 μM) or vehicle for 3 h (*$p < 0.05$ vs. LacZ/Vehicle, #$p < 0.05$ vs. LacZ/PE, $n = 5$). **c** NRCMs transduced with indicated adenovirus were treated with phenylephrine (PE, 100 μM) or vehicle for 6 h. Immunoblots of cell lysates (left) and statistical analyses of densitomeric measurements of p-p70 S6K (Thr 389) and p-mTOR (Ser 2448) (right) are shown (*$p < 0.05$ vs. sh-con/LacZ/Vehicle, #$p < 0.05$ vs. sh-con/LacZ/PE, &$p < 0.05$ vs. sh-con/KLF15/PE, $n = 4$). SE for short exposure. **d**, **e** NRCMs transduced with indicated adenovirus were incubated with DMEM containing either glucose or non-glucose substrates for one hour and then treated with phenylephrine (PE, 100 μM) or vehicle. **d** Immunoblots of cell lysates (left) and statistical analyses of densitomeric measurements of p-p70 S6K (Thr 389) and p-mTOR (Ser 2448) (right) are shown (*$p < 0.05$ vs. sh-con/Glucose/Vehicle, #$p < 0.05$ vs. sh-con/Glucose/PE, &$p < 0.05$ vs. sh-con/Lactate/PE, $n = 4$). **e** 48 h after PE treatment, myocytes were fixed and stained with anti-Troponin T. Cell surface area in each group was quantified and expressed relative to the control (*$p < 0.05$ vs. sh-con/Glucose/Vehicle, #$p < 0.05$ vs. sh-con/Glucose/PE, &$p < 0.05$ vs. sh-con/Lactate/PE, $n = 4$). Scale bar, 25 μm. Data shown as mean ± s.e.m. $P$ values were determined using one-way ANOVA followed by Newman-Keuls comparison test (**a**, **b**, **c**, **d**) or Kruskal–Wallis test followed by Dunn's comparison test (**e**)

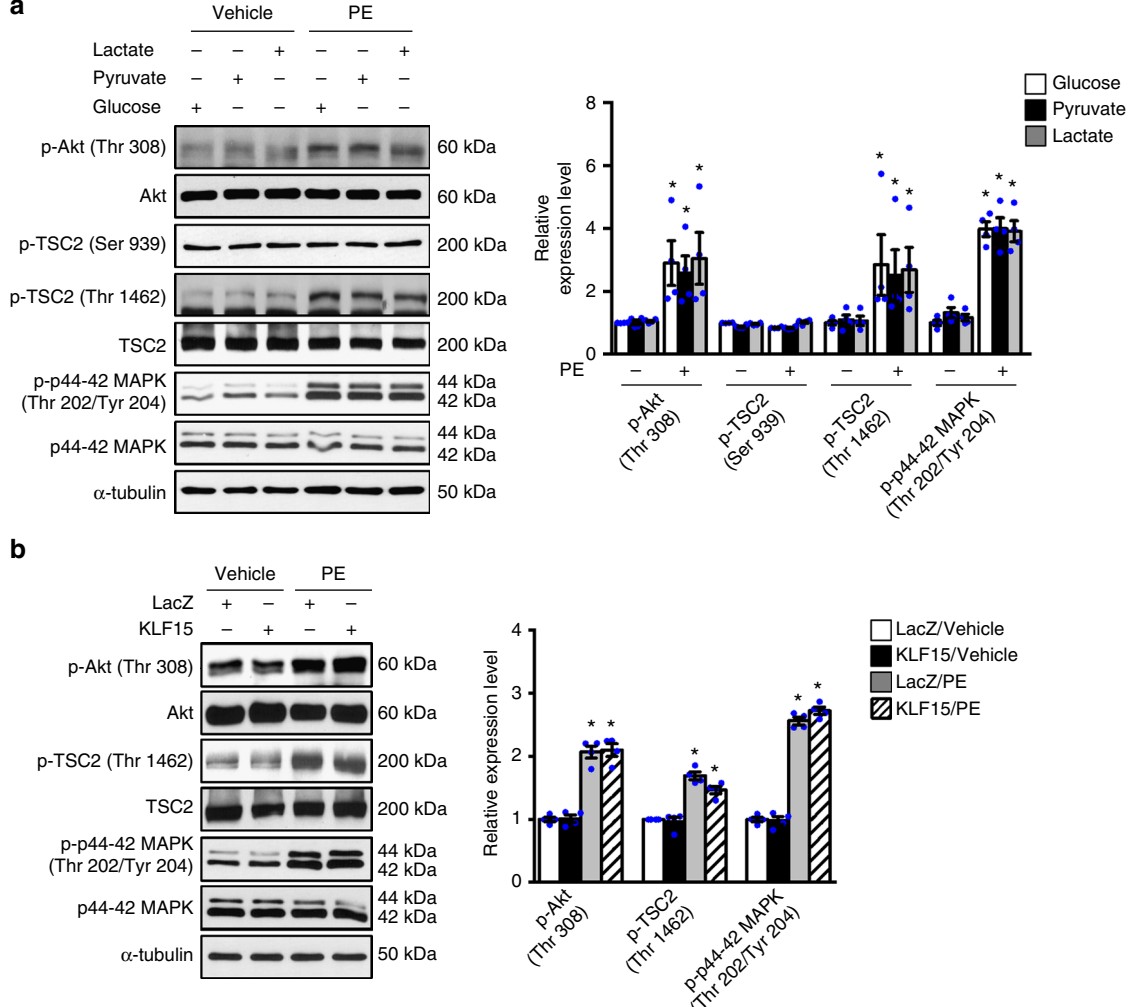

**Fig. 7** Glucose does not affect signaling upstream of mTOR. **a** NRCMs incubated with DMEM medium containing glucose or non-glucose substrates were treated with phenylephrine (PE, 100 µM) or vehicle for 6 h. Immunoblots of whole cell lysates (left) and statistical analyses of densitomeric measurements of p-Akt (Thr 308), p-TSC2 (Ser 939), p-TSC2 (Thr 1462), and p-p44–42 MAPK (Thr 202/Tyr 204) (right) are shown (*$p < 0.05$ vs. Glucose/PE(−), $n = 4$). **b** NRCMs transduced with indicated adenovirus were treated with phenylephrine (PE, 100 µM) or vehicle for 6 h. Immunoblots of whole cell lysates (left) and statistical analyses of densitomeric measurements of p-Akt (Thr 308), p-TSC2 (Thr 1462), and p-p44–42 MAPK (Thr 202/Tyr 204) (right) are shown (*$p < 0.05$ vs. LacZ/Vehicle, $n = 4$). Data shown as mean ± s.e.m. $P$ values were determined using one-way ANOVA followed by Newman–Keuls comparison test (**a**, **b**) or Kruskal–Wallis test followed by Dunn's comparison test (**a**)

The identification of the glucose-KLF15-BCAA regulatory circuit has potential implications in the cardiac remodeling of patients with diabetes or metabolic disorders. BCAAs catabolic enzyme gene expression is reduced in individuals with the metabolic syndrome and obesity[50,51]. Metabolomics studies has demonstrated a positive correlation between circulating BCAA levels and insulin resistance in obesity, and a high blood BCAA level is a risk factor for future development of diabetes[52,53]. These observations suggest a potential feedback regulation between glucose and BCAA metabolism[54]. A further understanding of the molecular mechanisms holds promise of revealing novel therapeutic targets for metabolic disorders.

In conclusion, our study demonstrates that glucose negatively regulates BCAAs degradation pathway through a KLF15 dependent mechanism. Mobilization of this pathway is required for mTOR activation and the development of cardiac hypertrophy. Thus, the study uncovers a molecular link between the metabolic programming and the growth signaling in the post-mitotic cells. Importantly, identification of the glucose and BCAA circuit adds

to the fundamental principle of metabolic regulation, which has significant implications beyond cell growth.

## Methods

**Animal model.** A transgenic mouse expressing human Glut1 driving by the alpha-myosin heavy chain promoter (designated as Glut1-TG) was generated on an FVB background[17]. PP2Cm systematic knock out mice (designated as PP2Cm KO) was achieved by gene targeting and maintained on C57BL/6 background[55]. All the mice were housed at 22 °C with a 12-h light, 12-h dark cycle with free access to water and standard chow. The experiments included in this study were performed with male mice. All protocols concerning animal use were approved by the Institutional Animal Care and Use Committee at University of Washington.

**Transverse aortic constriction (TAC) surgery.** PP2Cm KO mice and their littermate controls at the age of 10–12 weeks underwent TAC or sham surgery[56]. Briefly, mice were anesthetized with an intraperitoneal injection of 130 mg/kg ketamine and 8.8 mg/kg xylazine in saline. Mice were intubated with 20 G cannula and ventilated 140 breaths per minute by small animal TOPO ventilator (Kent Scientific). The aortic arch was exposed via a left thoracotomy and by carefully separating the thymus. A constriction of the transverse aorta was generated by tying a 6-0 Ethilon ligature against a 27-gauge blunt needle around the aorta between the brachiocephalic and left common carotid arteries. Promptly the needle

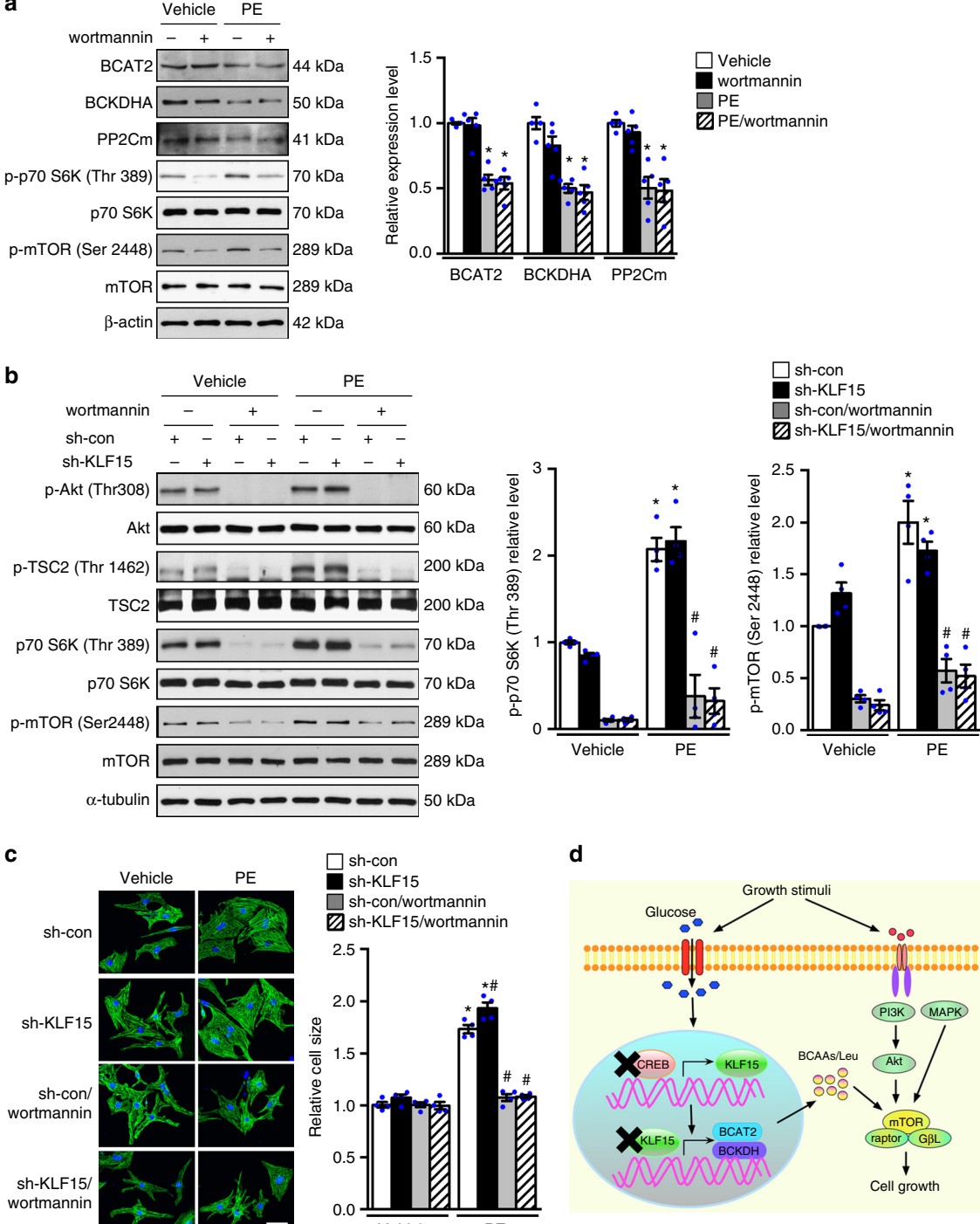

**Fig. 8** Activation of mTOR requires both growth stimulus and metabolic checkpoint. **a** NRCMs with preincubation of wortmannin (2 μM, 1 h) or vehicle were treated with phenylephrine (PE, 100 μM) or vehicle for 6 h. Immunoblots of total cell lysates (left) and statistical analyses of densitomeric measurements of BCAT2, BCKDHA, and PP2Cm (right) are shown (*p < 0.05 vs. Vehicle, n = 5). **b**, **c** NRCMs transduced with indicated adenovirus were preincubated with 2 μM wortmannin for 1 h and subsequently treated with phenylephrine (PE, 100 μM) or vehicle. **b** Immunoblots of total cell lysates (left) and statistical analyses of densitomeric measurements of p-p70 S6K (Thr 389) and p-mTOR (Ser 2448) (right) are shown (*p < 0.05 vs. sh-con/Vehicle, #p < 0.05 vs. sh-con/PE, n = 4). **c** 48 h after PE treatment, myocytes were fixed and stained with anti-Troponin T. Cell surface area in each group was quantified and expressed relative to the control (*p < 0.05 vs. sh-con/Vehicle, #p < 0.05 vs. sh-con/PE, n = 4). Scale bar, 25 μm. **d** Schematic illustration of the working hypothesis on how glucose regulates cell growth through modulating KLF15 mediated transcriptional control of BCAA degradation. Data shown as mean ± s.e.m. P values were determined using one-way ANOVA followed by Newman-Keuls comparison test (**a**, **b**) or Kruskal–Wallis test followed by Dunn's comparison test (**c**)

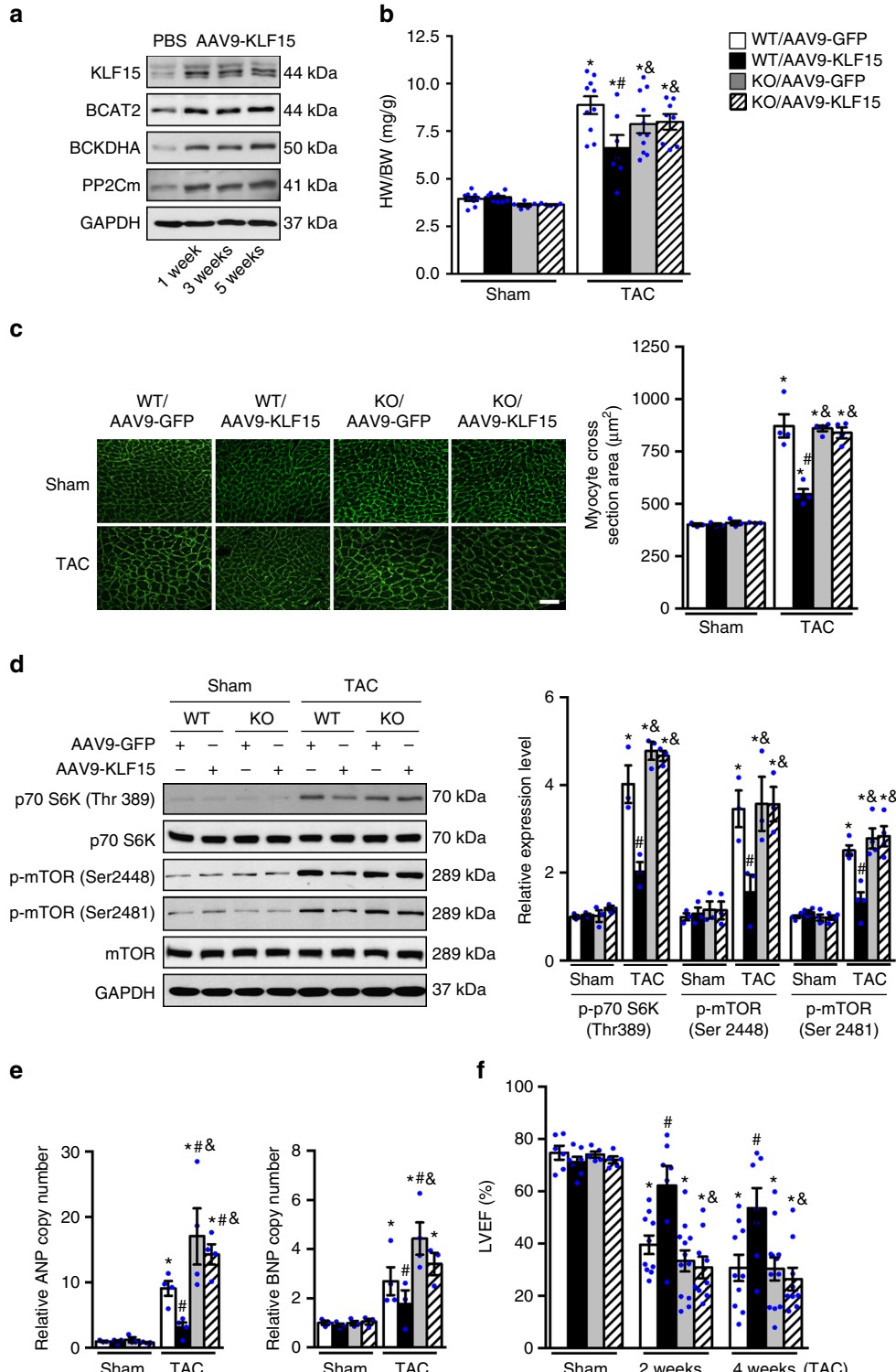

was removed and the chest and skin were closed by 5-0 polypropylene suture. The animal was removed from ventilation and kept on a heating blanket during recovery from anesthesia. SR buprenorphine (0.05 mg/kg) was administered subcutaneously for analgesia. Sham operated mice underwent all the same procedures as TAC mice excluding the constriction of the aorta. All mice were monitored every 12 h during the first 72 h post-surgery, followed by daily visits over the next 4 weeks.

**Transthoracic echocardiography**. The mice were anesthetized and maintained with 1–2% isofluorane in 95% oxygen. Trans-thoracic echocardiography was conducted at 2 and 4 weeks post TAC surgery with Vevo 2100 high-frequency,

high-resolution digital imaging system (VisualSonics) equipped with a MS400 MicroScan Transducer. A parasternal short axis view was used to obtain M-mode images for analysis of fractional shortening, ejection fraction, and other cardiac functional parameters.

**Bioinformatics analysis**. Glut1-TG and WT hearts were subjected to microarray analysis (Affymetrix). The preselected genes (Fold change > 1.25, $p < 0.05$ vs. WT) were analyzed using the Database for Annotation, Visualization, and Integrated Discovery (DAVID) (http://david.abcc.ncifcrf.gov)[57,58]. The lists of genes showing either down-regulation (806 genes) or up-regulation (1248 genes) in GLUT1-TG hearts were separately entered into the DAVID and subjected to Functional

**Fig. 9** Glucose and KLF15 mediated BCAA degradation is essential for cardiomyocyte growth in vivo. **a** Immunoblots of cardiac tissue homogenates from WT mice injected with AAV9-KLF15 or PBS for indicated time period are shown. **b–f** *PP2Cm* KO and WT mice were subjected to TAC surgery one week after retro-orbital injection of AAV9-KLF15 or control virus (AAV9-GFP). **b** The heart weight/body weight (HW/BW) ratio of *PP2Cm* KO and WT hearts with indicated AAV injection 4 weeks after TAC or sham operation (*$p < 0.05$ vs. WT/AA9-GFP/sham, #$p < 0.05$ vs. WT/AAV9-GFP/TAC, &$p < 0.05$ vs. WT/AAV9-KLF15/TAC, $n = 5$–11). **c** Representative wheat germ agglutinin staining and quantification of cardiomyocyte cross-sectional area in indicated hearts 4 weeks after TAC or sham operation (*$p < 0.05$ vs. WT/AA9-GFP/sham, #$p < 0.05$ vs. WT/AAV9-GFP/TAC, &$p < 0.05$ vs. WT/AAV9-KLF15/TAC, $n = 3$–4). Scale bar, 50 μm. **d** Left ventricles from indicated hearts were collected 3 days post surgery. Immunoblots of tissue homogenates (left) and statistical analyses of densitomeric measurements of p-p70 S6K (Thr 389), p-mTOR (Ser 2448) and p-mTOR (Ser 2481) (right) are shown (*$p < 0.05$ vs. WT/AA9-GFP/sham, #$p < 0.05$ vs. WT/AAV9-GFP/TAC, &$p < 0.05$ vs. WT/AAV9-KLF15/TAC, $n = 3$). **e** qRT-PCR measurements of *ANP* and *BNP* levels in *PP2Cm* KO and WT hearts with indicated AAV injection 4 weeks after surgery (*$p < 0.05$ vs. WT/AA9-GFP/sham, #$p < 0.05$ vs. WT/AAV9-GFP/TAC, &$p < 0.05$ vs. WT/AAV9-KLF15/TAC, $n = 3$–4). **f** Left ventricular ejection fraction (LVEF%) assessed by echocardiography at 2 weeks and 4 weeks post TAC surgery (*$p < 0.05$ vs. WT/AA9-GFP/sham, #$p < 0.05$ vs. WT/AAV9-GFP/TAC, &$p < 0.05$ vs. WT/AAV9-KLF15/TAC, $n = 5$–12). Data shown as mean ± s.e.m. *P* values were determined using one-way ANOVA followed by Newman–Keuls comparison test (**b**, **c**, **d**, **e**, **f**)

Annotation Chart analysis for general GO enrichment analysis and pathway enrichment analysis (Threshold: Count of 5 and EASE score of 0.05) using the KEGG pathway database[26]. KEGG pathway tool was utilized through DAVID online tools to visually map down-regulated genes involved in BCAAs degradation pathway in GLUT1-TG heart.

**Immunoblot analysis**. Heart homogenates or cell lysates were prepared using RIPA buffer containing 50 mM Tris (pH 7.4), 150 mM NaCl, 1% Triton X-100, 0.1% SDS, 1% deoxycholic acid, 1 mM EDTA, 1 mM $Na_3VO_4$, 1 mM NaF, 0.5 mM 4-(2-aminoethyl) benzenesulfonyl fluoride hydrochloride, 0.5 μg/ml aprotinin, and 0.5 μg/ml leupeptin. Equal amounts of proteins (10–20 μg) were subjected to SDS-PAGE. The nuclear and cytosolic fractions were prepared with NE-PER Nuclear and Cytoplasmic Extraction Reagents (Pierce). After proteins were transferred to a PVDF membrane, immunoblots were probed with the indicated antibodies. The protein abundance was analyzed densitometrically and normalized by the level of α-tubulin, β-actin, GAPDH or Lamin A. Antibodies used for immunoblots were purchased from the indicated companies. Primary antibodies against BCKDHA (1: 10000, ab138460), KLF15 (1: 2000, ab185958), p-mTOR (Ser 2481) (1: 10000, ab137133) were from Abcam. Primary antibodies against p-AMPKα (Thr 172) (1:1000, 2535), AMPKα (1: 2000, 2532), p-Akt (Thr 308) (1: 2000, 13038), Akt (1:5000, 4685), p-CREB (Ser 133) (1: 1000, 9198), CREB (1: 2000, 9197), GAPDH (1: 10000, 5174), p-mTOR (Ser 2448) (1: 5000, 5536), mTOR (1: 4000, 2983), p-p44–42 MAPK (Thr 202/204) (1: 10000, 4370), p44–42 MAPK (1: 5000, 4695), p-p70 S6K (Thr 389) (1: 1000, 9234), p70 S6K (1: 5000, 2708), p-TSC2 (Ser 939) (1: 5000, 3615), p-TSC2 (Thr 1462) (1: 1000, 3617), TSC2 (1: 4000, 4308), β-actin (1:5000, 4970) were from Cell Signaling Technology. Primary antibody against GFP (1: 10000, A-11122) was from Life Technology. Primary antibody against Glut1 (1: 2000, MABS132) was from Millipore. Primary antibody against BCKDK (1:1000, NBP2-15553) were from Novus Biologicals. Primary antibodies against BCAT2 (1:1000, 16417-1-AP) and PP2Cm (1: 2000, 14573-1-AP) were from Proteintech. Primary antibodies against Lamin A (1: 10000, sc-56137) and LDH (1: 5000, sc-33781) were from Santa Cruz Biotechnology. Primary antibodies against α-tubulin (1: 5000, T6199) and FLAG (1: 5000, F1804) were from Sigma. Uncropped images of immunoblots in the main figures are presented in the Supplementary Fig. 7.

**Cell cultures**. H9C2 and HEK 293 cells were purchased from American Type Culture Collection (ATCC) and were cultured at 37 °C in Dulbecco's modified Eagle's medium (DMEM) containing 5.5 mM Glucose (Life Technology) with 10% FBS. Cell lines used in this study were not verified as mycoplasma-negative and their morphology and growth characteristics were compared to published information to ensure their authenticity. Primary cultures of ventricular cardiac myocytes were prepared from 1-day-old Crl: (WI) BR-Wistar rats (Harlan Laboratories). A cardiac myocyte-rich fraction was obtained by centrifugation through a discontinuous Percoll gradient. Cells were cultured in complete medium (CM) containing DMEM/F-12 supplemented with 5% horse serum, 4 μg/ml transferrin, 0.7 ng/ml sodium selenite (Life Technologies, Inc.), 2 g/L bovine serum albumin (fraction V), 3 mM pyruvic acid, 15 mM HEPES, 100 μM ascorbic acid, 100 μg/ml ampicillin, 5 μg/ml linoleic acid and 100 μM 5-bromo-2′-deoxyuridine (Sigma). Myocytes were switched to serum-free medium for 24 h before any experiment. Primary adult cardiomyocytes were isolated from the heart of Sprague Dawley rats (Harlan Laboratories) using standard enzymatic technique[59]. Briefly, the heart was quickly removed, cannulated via the ascending aorta, and mounted on a modified Langendorff perfusion system and subsequently perfused with oxygenated Krebs–Henseleit Buffer (KHB) solution supplemented with collagenase II (Worthington) and hyaluronidase (Sigma) at 37 °C. Rod shaped adult cardiac myocytes were collected and cultured in serum-free M199 medium (Sigma) supplemented with 10 mM glutathione, 26.2 mM sodium bicarbonate, 0.02% bovine serum albumin and 50 U ml$^{-1}$ penicillin/streptomycin. All the experiments were conducted under 5.5 mM glucose concentration unless it's specified. Non-glucose substrates (12 mM pyruvate, 12 mM lactate or 6 mM lactate plus 6 mM acetate)

were added to DMEM containing 0 mM glucose for the indicated experiments. In order to maintain the same osmolality, D-mannitol (Sigma) was added to the medium samples if different concentrations of glucose were applied.

**Measurements of metabolite content and cell viability**. Intracellular ATP contents were measured using an ATP Bioluminescent Assay Kit (Sigma) according to the manufacturer's protocol. BCAA contents of mouse hearts or cells were measured using a BCAA Colorimetric Assay Kit (Sigma) according to the manufacturer's instructions. The BCAA content was normalized by protein levels. Viability of the cells was measured by CellTiter-Blue (CTB) assays (Promega) according to the supplier's protocol. In brief, cardiac myocytes ($1 \times 10^5$ per 100 μl) were seeded onto 96-well plates. On second day, the cells were changed to serum free medium for 24 h followed by incubation with either DMEM or glucose-free medium containing non-glucose substrates for 48 h before the CTB assay.

**Construction of adenoviral expression vectors**. Briefly, pBHGloxΔE1,3Cre (Microbix), including the ΔE1 adenoviral genome, was co-transfected with pDC shuttle vector containing the gene of interest into 293 cells using Lipofectamine 2000 (Invitrogen). Through homologous recombination, the test genes were integrated into the E1-deleted adenoviral genome. The viruses were propagated in 293 cells as described[60]. The cDNA clones for Glut1 and KLF15 were purchased from Origene. Adenovirus harboring β-galactosidase (Ad-LacZ) was used as a control.

**Construction of shRNA adenoviral expression vectors**. Adenoviruses harboring short hairpin RNA (shRNA) for Glut1 (Ad-sh-Glut1), KLF15 (Ad-sh-KLF15), BCAT2 (Ad-sh-BCAT2), and PP2Cm (Ad-sh-PP2Cm) were generated using the following hairpin forming oligos:

Glut1:(5′-GCTTATGGGTTTCTCCAAACTTTCAAGAGAAGTTTGGAGAAACCCATAAGCTTTTTT-3′)

KLF15:(5′-GCAAGACAAATGGAGCCATATTTCAAGAGAATATGGCTCCATTTGTCTTGCTTTTTT-3′)

BCAT2:(5′-GCAGAACGCAAGGTCACTATGTTCAAGAGACATAGTGACCTTGCGTTCTGCTTTTTT-3′)

PP2Cm:(5′- GCTATACTTTGCAGTCTATGATTCAAGAGATCATAGACTGCAAAGTATAGCTTTTTT-3′)

These oligos and their corresponding antisense oligos with ApaI and Hind III overhangs were synthesized, annealed, and subcloned into the pDC311 vector. The loop sequences are underlined. Recombinant adenoviruses were generated using homologous recombination in 293 cells as described above.

**Generation and administration of AAV (serotype 9)**. *KLF15* cDNA and GFP were separately cloned into ITR-containing AAV plasmid harboring the chicken cardiac TNT promoter to yield constructs pAAV9.cTnT-KLF15 and pAAV9.cTnT-GFP, respectively. AAV was packaged in 293 T cells using AAV9:Rep-Cap and pAd:deltaF6 (Penn Vector Core); then purified and concentrated by gradient centrifugation[61,62]. AAV9 titer was determined by Droplet Digital PCR (ddPCR). AAV9 virus ($1 \times 10^{12}$ virus genome per animal) was injected into *PP2Cm* KO mice and their control littermates by retro-orbital injection.

**Immunocytochemistry**. Neonatal cardiac myocytes in 4-well chambers were washed with phosphate-buffered saline (PBS) three times, fixed with 4% paraformaldehyde for 15 min, permeabilized in 0.3% Triton X-100 for 10 min and blocked with 10% normal goat serum for 1 h at room temperature. The following antibodies were used as primary antibodies: KLF15 (1: 200, ab185958, abcam) and Troponin T (1: 400, MS295P1, Thermo Scientific). Alexa Fluor 488 Dye-conjugated or Alexa Fluor 555 Dye-conjugated secondary antibody (Invitrogen) was used for detecting indirect fluorescence. Slides were mounted with Prolong Diamond Antifade Mountant containing DAPI (Life Technology).

**Single-cell fluorescent hexose uptake assay**. Neonatal cardiac myocytes in 4-well chambers were washed with phosphate-buffered saline (PBS) for three times. The myocytes were subsequently incubated with 2-(N-(7-Nitrobenz-2-oxa-1,3-diazol-4-yl)Amino)-2-Deoxyglucose (2-NBDG, 300 μM) (Life Technology) in Krebs buffer without glucose (145 mM NaCl, 5 mM KCl, 6 mM CaCl$_2$, 1 mM MgCl$_2$, 25 mM HEPES Na, and 10 mM NaHCO$_3$, pH 7.4) for 1 h at 37 °C in the dark. After rinsed with Krebs buffer, intracellular fluorescence was acquired by confocal microscope. 2-NBDG uptake was estimated by comparing intracellular fluorescence with the extracellular signal. The images were quantified by ImageJ software (NIH). Approximately 75–100 cells were analyzed for each condition per experiment. Glucose uptake was expressed as the fold change over the control cells.

**Cryosectioning and immunofluorescence of GFP**. To visualize transgene distribution after retro-orbital injection, hearts from mice injected with AAV9-GFP or PBS were harvested and fixed for four hours at 4 °C in PBS containing 0.1% glutaraldehyde/1.5% paraformaldehyde/20% sucrose, embedded in Tissue-Plus O.T.C Compound (Fisher Scientific) and quickly frozen in liquid nitrogen. Cryosections of 5 μm thickness were cut and GFP expression was detected by anti-GFP antibody (1: 400, A-11122, Invitrogen). Slides were mounted with Prolong Diamond Antifade Mountant containing DAPI (Life Technology).

**Targeted metabolite profiling**. Targeted reverse phase (RP) liquid chromatography (LC)-mass spectrometry (MS) metabolite analysis was performed at the University of Washington Northwest Metabolomics Research Center[63]. Briefly, frozen heart tissues (~20 mg) were homogenized in 2 mL 80:20 methanol:water. The soluble extracts were collected and then dried at 30 °C in a Speed-Vac. Dried extracts were reconstituted in 200 μL of 5 mM ammonium acetate in 95% water/5% acetonitrile + 0.3% acetic acid (pH = 3.5) and filtered prior to LC-MS analysis. Targeted LC-MS analysis was performed in RP chromatography mode on two parallel C18 analytical columns, one for positive and the other for negative MS ionization mode, respectively. LC system was composed of two Agilent 1260 binary pumps, an Agilent 1260 auto-sampler and an Agilent 1290 column compartment containing a column-switching valve (Agilent Technologies). The chromatography was performed in RP mode using solvents A (95% water/5% acetonitrile + 0.3% acetic acid) and B (5% water/95% acetonitrile + 0.3% acetic acid): 5% B for 2 min, 5% B to 80% B in 3 min, 80% B for 3 min, 80%B to 5% B in 3 min, and 5% B for 7 min (18 min total for each ionization mode). After the chromatographic separation, MS ionization and data acquisition was performed using AB Sciex QTrap 5500 mass spectrometer (AB Sciex) equipped with electrospray ionization (ESI) source. Multiplereaction-monitoring (MRM) mode was used for targeted data acquisition. Total 137 metabolites were measured as relative MRM peak areas using MultiQuant 2.1 software (AB Sciex).

**Construction of luciferase reporter vectors**. Four proximal 5′ regions (−1068, −541, −322, −189 bp relative to transcript start site, respectively) of mouse KLF15 promoter region were amplified by PCR. Promoter PCR product was cloned into a firefly luciferase reporter pGL3-Basic vector to drive luciferase expression (Promega). 3XCRE (TGACGTCA) was inserted into pGL3 vector (CRE-luc). A PCR product containing 638 bp upstream of the rat atrial natriuretic factor (ANF) transcription start site was cloned into pGL3 vector (ANF-luc)[64].

**Transient transfection and luciferase assays**. Neonatal cardiomyocytes were plated on 12-well plates and transfected with 1 μg of the indicated luciferase vectors with or without overexpression plasmids using Fugene 6. Six hours after transfection, the cells were washed and transduced with the indicated adenovirus. Then the cells were lysed with Passive Lysis Buffer, and the transcriptional activity was determined using a luciferase assay system (Promega). For each construct, more than three independent experiments were performed in triplicate.

**Chromatin immunoprecipitation (ChIP)**. Neonatal cardiomyocytes were treated with 1% formaldehyde for 10 min to cross-link proteins and chromatin. The reaction was stopped by adding 0.125 M Glycine for 5 min. Cells were washed with cold PBS twice. Cells were then resuspended in 1 ml ChIP lysis buffer (20 mM Tris-HCl (pH 8.0), 85 mM KCl, 0.5% NP-40) for 10 min at 4 °C and centrifuged at 2600×g to pellet the nuclei. The cell nuclei were resuspended in 400 μL nuclei lysis buffer (50 mM Tris-HCl (pH 8.0), 10 mM EDTA, 1% SDS) and then subjected to sonication 5 times for 30 s with 30 s intervals. Purified chromatin was analyzed on a 1% agarose gel to determine the shearing efficiency. As a control, normal IgG (sc-2027, Santa Cruz Biotechnology) was used as a replacement for the CREB antibody (sc-186×, Santa Cruz Biotechnology). The ChIP procedure was performed as in the supplier's protocol (Active Motif). The primer information is listed in Supplementary Table 3.

**Histological analyses**. Mouse heart tissues were rinsed with PBS and fixed in 10% neutral buffered formalin overnight. Fixed samples were dehydrated, embedded in paraffin wax and sectioned for Wheat Germ Agglutinin staining. The outline of cross-sectional areas for 100–150 myocytes was traced and quantified with ImageJ

software in each section. Suitable cross sections of the myocardium were defined as having nearly circular capillary profiles and circular-to-oval myocyte sections.

**RNA extraction and quantitative real-time PCR**. Total RNA was extracted from cells using Quick-RNA MiniPrep Kit (Zymo Research) and from frozen LV tissue using the RNeasy Fibrous Tissue Mini Kit (Qiagen) according to the manufacturer's instructions. Total RNA was reverse-transcribed into the first-strand cDNA using the Superscript First-Strand Synthesis Kit (Invitrogen). cDNA transcripts were quantified by Rotor Gene Real-Time PCR System (Qiagen) using SYBR Green (Biorad). Results of mRNA levels were normalized to 18S rRNA levels and reported as fold-change over control. The primer information is listed in Supplementary Table 3.

**Statistical analysis**. The numbers of independent experiments are specified in the relevant figure legends. Data are expressed as mean ± standard error of the mean (SEM). Statistical analysis was performed with Prism 7.0 software (GraphPad). Normal distribution of the data was assessed using Shapiro–Wilk normality test. Statistical comparisons between groups were conducted by unpaired Student's $t$-test or one-way ANOVA followed by a Newman–Keuls comparison test for normal distribution. Otherwise, Mann–Whitney test or Kruskal–Wallis test followed by Dunn's multiple comparison test were applied. (No assumption was made that variance was similar between the groups being statistically compared). The value of $p < 0.05$ was considered to be significant. No statistical method was used to predetermine samples size. There was a 10–15% surgical failure rate with the mouse TAC surgery, and it was pre-established that these mice would be excluded from the analysis. Animals within each cohort were randomly assigned to various groups based upon experimental design. Investigators were blinded for experiments including AAV injection, TAC surgery and echocardiography. Otherwise, the experiments were not randomized and investigators were not blinded to allocation during experiments and outcome assessments.

**Data availability**. Microarray data displayed in Fig. 1a, Supplementary Fig.1a, b and Supplementary Table 1 are deposited in NCBI Gene Expression Omnibus under accession code GSE110766. All other remaining data are available within the article and Supplementary Files, or available from the authors upon request.

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

## Acknowledgements

We thank members of the Tian laboratory for the support. This work was supported in part by U.S. National Institutes of Health Grants HL-088634, HL-118989, and HL-129510 (to R.T.), the American Heart Association Postdoctoral Fellowship 15POST21620006 (to D.S.), the American Heart Association Scientist Development Grant 14SDG18590020 (to S.C.K.) and the Deutsche Forschungsgemeinschaft Research Fellowship 2764/1-1 (to J.R.). We thank Dr. Yibin Wang (University of California, Los Angeles) for generously providing the PP2Cm KO mice. We thank Dr. Pete Watson (University of Colorado) for his courtesy of CREB adenovirus. We thank Dr. Karol Bomsztyk (University of Washington) for providing Diagenode Bioruptor for ChIP assays. We also thank Dr. Roger J. Hajjar (Icahn School of Medicine at Mount Sinai) for providing AAV.cTnT construct. We also appreciate and thank the NHLBI Gene Therapy Resource Program and the Penn Vector Core Facility for the production of AAV9 virus.

## Author contributions

D.S. and R.T. designed the experiments and wrote the manuscript. D.S., Z.Z., S.W.C., J.Y., and D.C. performed the experiments and D.S. analyzed the data. O.V. performed the AAV injection and mouse surgery. O.V. and J.R. performed the mouse echocardiography and D.S. analyzed the data. H.G., D.D., and D.R. performed the metabolomics analysis.

J.R. and S.C.K. provided advice on data analyses and the manuscript. R.T. supervised the project.

## Additional information

**Competing interests:** The authors declare no competing interests.

