## [Peer Review File · Nature Communications]

Reviewers' comments:

Reviewer #1 (Remarks to the Author):

This report by Shao et al characterized the impact of high glucose on branched-chain amino acid degradation. This report focused on cardiac hypertrophy as the biological outcome in the context of high glucose effect on bcaa catabolism. This is a highly relevant issue to the field as high BCAA level is a newly reported metabolic hallmark of failing hearts in human and disease models, while glucose metabolic dysregulation is also a common feature of the disease. More interestingly, BCAA catabolic activity is emerging as a key metabolic process in numerous diseases, including cancer, diabetes and metabolic disorders. Therefore, the insights about the functional interaction between glucose and bcaa may have general implications beyond cardiac diseases.

This report focused on characterizing the impact of high glucose on BCAA catabolism in cardiomyocytes and its role in cardiac hypertrophy. For that, the authors employed several genetic and gene-transfer tools in different in vitro and in vivo hypertrophy models. The data was consistent in supporting KLF-15 as a pivotal downstream transcription factor of high glucose mediated inhibition of BCAA catabolism in cardiomyocytes during hypertrophic response. Extensive mechanistic studies were also performed to implicate CREB mediated regulation of KLF-15 expression under high glucose signal. This is an interesting and potentially important finding to link glucose and BCAA metabolic process in the context of hypertrophic stimulation in cardiomyocytes. The mechanistic study was comprehensive and logically organized. The dissection of energetic vs. signaling effect of glucose was particularly important and well demonstrated. The effect of BCAA catabolism inhibition as an essential but not sufficient signal for the onset of cardiac hypertrophy was supported by both in vitro and in vivo evidence.

Several minor clarifications would be helpful as outlined below:

- 1). Authors imply that accumulated BCAA is the final outcome and necessary consequence for mTOR activation during hypertrophy, via a Glut1-CREB-KLF-15 mediated regulation pathway. However, under hypertrophic stimulation in vitro by PE, KLF-15 expression was consistently downregulated along with BCAA catabolic genes, but BCAA increase was not only modest but also very transient (less than 6 hours). It is unclear about the concentrations of BCAA in post-TAC heart under this study. Since most of the cultured media have high level of BCAA, it is also not clear if supplement of BCAA would be sufficient to restore cardiac hypertrophy blocked by Glut-1 knockdown or lower BCAA was sufficient to block hypertrophy. This is an important question about the role of BCAA and its catabolic activity in regulation of cellular growth and pathology. Authors should carefully demonstrate or discuss based on their data about this question.
- 2). The entire regulatory network was extensively tested by experimental data. It would serve the readers better if the outline of the overall regulatory scheme can be illustrated as a figure panel.

3). Authors did not discuss the differences between physiological growth vs. pathological growth in hearts, and whether the same scheme is implicated. Meanwhile, the discussion should also include potential implication of cardiac remodeling in the context of diabetes or glucose disorder, as well as potential implications how intervention of these pathways can be exploited as future therapies.

Reviewer #2 (Remarks to the Author):

This manuscript by Shao and colleagues answers a fundamental question that has existed in the field for decades – how are metabolic changes associated with hypertrophic growth, and is metabolism a driver or a “back seat passenger”. This manuscript demonstrates that increased glucose metabolism drives mTOR signalling and hypertrophy via KLF15 signalling and regulation of branched chain amino acid metabolism. This manuscript resolves many controversies in the field and unpicks a critical role for metabolism as a direct regulator of signalling and growth.

I have a number of minor changes to what is otherwise an excellent high-quality study:

1. Abstract – “although ATP provision from increased glucose reliance is dispensable during the hypertrophic growth of cardiomyocytes”. This statement cannot be confidently made based on work in this study. This sentence detracts from the key message in this paper and should be removed – there are far more important findings to focus on.
2. Results – page 5 line 3 “did not alter cardiac function or survival of the mice”. This is confusing given the title of the paper referenced “prevents development of heart failure”. This sentence is also irrelevant for everything that follows.
3. Fig 1 C – change “Citrate cycle” to TCA cycle that the authors use elsewhere
4. Methods – more details is needed for the specific cell culture expts – when 5mM and 25mM glucose are compared in earlier figures are these mannitol controlled experiment? How long is PE added for on these experiments? Fig 3 and 4 – are these at 25mM glucose, 12mM glucose or 5 mM glucose?
5. Statistics. The authors state “no statistical method was used to predetermine sample size”. How then do the authors justify only doing n=3 or 4 for some of their cell experiments? This is a very low n number for this work, particularly in light of the impact factor of the journal this is submitted to. For cell culture experiments I believe an n=3 or 4 is insufficient for confidence in data if power calculations have not been carried out to justify this low number.
6. Fig 2 L – BCAT2 blot poor quality – is there a reason for this?
7. Fig 3 – issues with statistics. By eye it looks like some groups are significantly different to each other but haven’t been denoted as such on the graphs. Fig3D – it appears vehicle sh-KLF15 is

significantly different to vehicle sh-con? This would mean that KLF15 has a regulatory role even without PE? Similarly Fig3E KFL15 veh vs lacZ veh look significantly different?

8. Fig 3F and last line of page 8. The authors state “the anti-hypertrophic effect of KLF15 overexpression was completely abolished”. The KLF15/sh-PP2Cm PE treated group looks significantly different to sh-con PE and sh-PP2Cm PE bars – both this isn’t shown on the cell size graph. I would argue that this group sits partly between these groups and KLF15 PE, therefore, completely abolished is not an accurate description of this data.

9. Page 9 and S4F – why hasn’t AMPK phosphorylation been quantified as the other targets have?

10. Fig 4 F, G and B – The authors show that AMPK is activated by lactate in neonatal but not adult cardiomyocytes (4B vs. S4G). Given the neonatal obviously have this stress response and mTOR is sensitive to energy status – were neonatal or adult cell used in Fig 4F and 4G. If this data is neonatal – then this data is not suitable as you can’t delineate the stress response from this mTOR effect. If neonatal – you would need to repeat this with lactate+acetete where neonatal cells don’t have the AMPK response. If it’s adult cells that were used to avoid this issue, then please make this clearer in the manuscript.

11. Bottom of page 11 – Fig 5D and E. This was really difficult to follow regarding the input from both arms and despite rereading multiple times I still couldn’t see how the data supported this. Please address this by making clearer or reinforcing with further experiments if needed.

12. Fig 6 – are there any measures that surgical severity was the same between all groups, as the WT/AAV9-KLF15 data could be explained if the surgery was less severe.

Reviewer #3 (Remarks to the Author):

This is an exciting set of observations that glucose metabolism and the BCAA degradation pathway through a KLF15-dependent transcriptional mechanism to mechanisms of cell growth. The study is well written, well-presented and advanced novel and intriguing findings. However, the study could be strengthened by addressing some of the issues outlined below:

(1) Pathological cardiac hypertrophy induced by PE or TAC is not normal cell growth. The title “Glucose Promotes Cell Growth...” is an overstatement and should be modified.

(2) The link between increased intracellular glucose promoting cell growth through a KLF15-BCAA mechanism needs to be solidified. Does overexpression of GLUT1 or high glucose medium sufficient to induce hypertrophy under basal conditions or after PE?

(3) BCAA accumulation in the heart due to its reduced degradation appears to be detrimental to the heart, as suggested by Sun's 2016 study. In current study, KLF15's rescue is dependent on PP2Cm-BCAA degradation pathway also suggested a detrimental role of BCAA accumulation. Furthermore, KLF15 is known to be reduced in many heart disease states. This leads the reviewer to wonder why/how the Glut1-TG heart, which exhibited reduction in both KLF15 and BCAA degradation pathway, was healthy at baseline and protected from TAC? The authors should discuss.

(4) "Suppression of BCAA Degradation by Glucose Is Required for Maintaining mTOR Activity during Cardiac Hypertrophy". The claim is not well supported. The authors showed PE-induced mTOR activation requires glucose or BCAA accumulation. But there is no data supporting intracellular glucose level and BCAA accumulation is connected/related after PE treatment or PE needs both to activate mTOR. In addition, it has been reported that BCAA, such as Leu, could directly activate mTOR.

Point-by-point replies to the reviewers' comments

We would like to thank the editor and the reviewers for their careful review, constructive criticisms and thoughtful suggestions. To address the questions raised by the editor and reviewers, we have performed additional experiments and thoroughly revised the manuscript. Our point-to-point responses are included below.

Reviewer #1:

We thank the reviewer for the enthusiastic support of our study and appreciate the questions raised which helped us to improve the manuscript. Our point-to-point responses are as follows:

1). Authors imply that accumulated BCAA is the final outcome and necessary consequence for mTOR activation during hypertrophy, via a Glut1-CREB-KLF-15 mediated regulation pathway. However, under hypertrophic stimulation in vitro by PE, KLF-15 expression was consistently downregulated along with BCAA catabolic genes, but BCAA increase was not only modest but also very transient (less than 6 hours). It is unclear about the concentrations of BCAA in post-TAC heart under this study. Since most of the cultured media have high level of BCAA, it is also not clear if supplement of BCAA would be sufficient to restore cardiac hypertrophy blocked by Glut-1 knockdown or lower BCAA was sufficient to block hypertrophy. This is an important question about the role of BCAA and its catabolic activity in regulation of cellular growth and pathology. Authors should carefully demonstrate or discuss based on their data about this question.

We agree with the reviewer that manipulating the BCAA concentration in culture medium is not a good approach. BCAA level in the medium is several folds higher than blood levels. It is difficult to influence intracellular BCAA content by further increasing [BCAA] in the medium. Thus, we chose to manipulate BCAAs catabolism using KLF15 overexpression or knockdown strategy to disrupt BCAAs catabolism (Fig. 3c, d; Fig 4d, e). Similarly, knocking down BCAT2 or PP2Cm restored mTOR activation and cell growth when glucose was withdrawn (Fig. 4f, g; Supplemental Fig. 4m-p).

The transient accumulation of BCAA in cultured cell is consistent with the observation in TAC and MI hearts in which the increase of cardiac BCAA peaks at one week after the stimulus (Sansbury et al Circulation HF 2014, Wang et al AJP-heart circulatory physiology 2016). Although the BCAA catabolism pathway remained downregulated, BCAA accumulation was not robust in later stage of hypertrophy and heart failure (Sansbury et al Circulation HF 2014, Wang et al AJP-heart circulatory physiology 2016, Sun et al Circulation 2016, Lai et al, Circulation HF 2014). We speculate that once cell growth is triggered, increased protein synthesis will prevent further accumulation of BCAA. These points are now included in the revised manuscript (page 10 line 16-19).

2). The entire regulatory network was extensively tested by experimental data. It would serve the readers better if the outline of the overall regulatory scheme can be illustrated as a figure panel.

Thank you for the suggestion. We have added a scheme to summarize the entire study (Fig. 5f, page 12 line 4).

3). Authors did not discuss the differences between physiological growth vs. pathological growth in hearts, and whether the same scheme is implicated. Meanwhile, the discussion should also include potential implication of cardiac remodeling in the context of diabetes or glucose disorder, as well as potential implications how intervention of these pathways can be exploited as future therapies.

We thank the reviewer for raising a critical point here. We have performed new experiments to determine if increased glucose uptake by physiological stimulus for growth such as IGF-1 or insulin alters this pathway. We found that the expression of KLF15 and BCAA degradation enzymes were downregulated *in vitro* and *in vivo* during IGF-1 and insulin treatment respectively (Supplementary Fig. 3e-g, page 8 line 15-17). We now include a discussion about the implication of our findings in metabolic disorders (page 17 line 13-21).

Supplementary Figure 3

(e) Immunoblots of nuclear fractions from NRCMs treated with IGF-1(10 nM) or vehicle (upper) and statistical analysis of densitometric measurement of nuclear KLF15 (lower) are shown (*p<0.05 vs. vehicle, n=3). (f) Immunoblots of cell lysates from NRCMs treated with IGF1 (10 nM) or vehicle (left) and statistical analyses of densitometric measurements of BCAT2, BCKDHA and PP2Cm (right) are shown (*p<0.05 vs. vehicle, n=3). (g) WT mice were subjected to 18 hour fasting and then injected with vehicle or insulin (1 Unit per kg). Heart tissues were collected 2 hours after injection. qRT-PCR analysis of indicated genes were examined. The expression was normalized to 18S rRNA and reported as fold change over vehicle (*p<0.05 vs. vehicle, n=5).

Reviewer #2:

We thank the reviewer the insightful review and appreciate the comment that our study “resolves many controversies in the field and unpicks a critical role for metabolism as a direct regulator of signaling and growth”. All the points raised by the reviewer is addressed as follows:

1. Abstract – “although ATP provision from increased glucose reliance is dispensable during the hypertrophic growth of cardiomyocytes”. This statement cannot be confidently made based on work in this study. This sentence detracts from the key message in this paper and should be removed – there are far more important findings to focus on.

We have removed the sentence from the abstract as suggested by the reviewer.

2. Results – page 5 line 3 “did not alter cardiac function or survival of the mice”. This is confusing given the title of the paper referenced “prevents development of heart failure”. This sentence is also irrelevant for everything that follows.

The sentence is to describe the cardiac phenotype of Glut1-TG under baseline condition (without stress) at young and old ages. Indeed, Glut1-TG prevented the development of heart failure under pressure overload condition. We are sorry for the confusion. We have revised the sentence for clarity (page 5 line 5).

3. Fig 1 C – change “Citrate cycle” to TCA cycle that the authors use elsewhere

Yes, we have made the change.

4. Methods – more details are needed for the specific cell culture expts – when 5mM and 25mM glucose are compared in earlier figures are these mannitol controlled experiments? How long is PE added for on these experiments? Fig 3 and 4 – are these at 25mM glucose, 12mM glucose or 5 mM glucose?

When 5mM and 25mM glucose are compared, we have added mannitol in 5 mM glucose culture medium. For Figure 3 and 4, all the experiments were conducted under 5 mM glucose concentration, unless it's specified. The time for PE treatment were varied in each set of experiments, which were specified in the respective Figure Legends. We also added more details in the method section as the reviewer suggested (page 22 line 20-24).

5. Statistics. The authors state “no statistical method was used to predetermine sample size”. How then do the authors justify only doing n=3 or 4 for some of their cell experiments? This is a very low n number for this work, particularly in light of the impact factor of the journal this is submitted to. For cell culture experiments I believe an n=3 or 4 is insufficient for confidence in data if power calculations have not been carried out to justify this low number.

We agree that n=3 is small sample size. Therefore, we have repeated some of the experiments and have now eliminated n=3 for cell experiments in the main figures. We have also re-performed the statistical analysis. Specifically, we determined the distribution of the data using prism software. For those that did not present a normal distribution, we added nonparametric tests for statistical analysis. In this case, we have used the Mann-Whitney test for two group comparison or Kruskal-Wallis test followed by Dunn's multiple comparison test for more than two groups. Our conclusion was not changed by increasing the N or additional statistical analysis. We have expanded the method section accordingly (page 29 line 3-8).

6. Fig 2 L – BCAT2 blot poor quality – is there a reason for this?

Thank you. We have replaced this blot with one of better quality.

7. Fig 3 – issues with statistics. By eye it looks like some groups are significantly different to each other but haven't been denoted as such on the graphs. Fig3D – it appears vehicle sh-KLF15 is significantly different to vehicle sh-con? This would mean that KLF15 has a regulatory role even without PE? Similarly, Fig3E KLF15 veh vs lacZ veh look significantly different?

Good catch. We have re-examined the statistical analysis and included denotations for all comparisons with statistical significance in Figure 3d and 3e.

8. Fig 3F and last line of page 8. The authors state “the anti-hypertrophic effect of KLF15 overexpression was completely abolished”. The KLF15/sh-PP2Cm PE treated group looks significantly different to sh-con PE and sh-PP2Cm PE bars – both this isn’t shown on the cell size graph. I would argue that this group sits partly between these groups and KLF15 PE, therefore, completely abolished is not an accurate description of this data.

We agree and have re-written the sentence more accurately (page 9 line 4).

9. Page 9 and S4F – why hasn’t AMPK phosphorylation been quantified as the other targets have?

The reviewer raised a good point. We have already quantified the AMPK phosphorylation in neonatal and adult rat cardiomyocytes cultured with glucose and non-glucose medium (Fig. 4b and Supplementary Fig. 4g, respectively). As the reviewer requested, we have now included the quantification of AMPK phosphorylation in Supplementary Fig. 4f.

10. Fig 4 F, G and B – The authors show that AMPK is activated by lactate in neonatal but not adult cardiomyocytes (4B vs. S4G). Given the neonatal obviously have this stress response and mTOR is sensitive to energy status – were neonatal or adult cell used in Fig 4F and 4G. If this data is neonatal – then this data is not suitable as you can’t delineate the stress response from this mTOR effect. If neonatal – you would need to repeat this with lactate+acetate where neonatal cells don’t have the AMPK response. If it’s adult cells that were used to avoid this issue, then please make this clearer in the manuscript.

We are sorry for the confusion. In Figure 4f and 4g, those cells are neonatal cardiomyocyte, which has been specified in the Figure Legends (page 42 line 11). Although AMPK appeared to be activated in these cells when cultured with lactate, mTOR inhibition was also observed when cultured with lactate+acetate in which there was no AMPK activation (Figure 4b). This finding led us to conclude that mTOR inhibition upon glucose removal (using either lactate+acetate or lactate alone) is through AMPK independent mechanisms. When we performed the same experiment using adult cardiomyocyte cultured with lactate (no AMPK activation), we obtained similar results as using neonatal cardiomyocyte (Fig. 4f, g vs. Supplementary Fig. 4o, p). We have now made the reasoning clear in the revised manuscript (page 9 line 16, line 19-20).

11. Bottom of page 11 – Fig 5D and E. This was really difficult to follow regarding the input from both arms and despite rereading multiple times I still couldn’t see how the data supported this. Please address this by making clearer or reinforcing with further experiments if needed.

We have revised this part to improve clarity (page 12 line 1-2). In addition, a scheme is included to summarize the overall findings (Fig. 5f, page 12, line 4), which will hopefully help the readers.

12. Fig 6 – are there any measures that surgical severity was the same between all groups, as the WT/AAV9-KLF15 data could be explained if the surgery was less severe.

It is known that pressure overload induced by transverse aortic constriction (TAC) has variations. However, our in-house surgery team has been consistently generating a pressure gradient of 40-50mmHg (Kolwicz SC Jr et al. *Cir Res*, 2012; Garcia-Menendez L et al. *Am J Physiol Heart Circ Physiol*, 2013). Although we did not measure the pressure gradient across the constriction in this cohort of mice, it is important to note that AAV injection, TAC surgery and echocardiographic measurements in this study were performed in a blinded fashion to avoid any selective bias (page 29 line 13-15). Moreover, the number of mice used for the experiments are sufficiently large (10-12), which also help to minimize the influence of surgical variations.

Reviewer #3:

We thank the reviewer for a thorough review and appreciate the comment that our study “study is well written, well-presented and advanced novel and intriguing findings”. We have followed the reviewer’s suggestions to address the issues outlined below:

1. Pathological cardiac hypertrophy induced by PE or TAC is not normal cell growth. The title “Glucose Promotes Cell Growth...” is an overstatement and should be modified.

We thank the reviewer for raising an important point here. Another reviewer also asked a similar question. Thus, we performed new experiments to determine if increased glucose uptake during physiological stimulation for growth such as IGF-1 or insulin alters the KLF15-BCAA degradation pathway. We found that the expression of KLF15 and BCAA degradation enzymes were downregulated *in vitro* and *in vivo* during IGF-1 and insulin treatment respectively (Supplementary Fig. 3e-g, page 8, line 15-17). Moreover, the downregulation of KLF15 and subsequent BCAA accumulation by high glucose is also observed in cell types other than cardiomyocytes (Fig. 1h; Supplementary Fig. 1g, h; Supplementary Fig. 2b, c). These findings suggest that glucose-KLF15-BCAA axis is involved in cell growth beyond pathological cardiac hypertrophy.

Supplementary Figure 3

(e) Immunoblots of nuclear fractions from NRCMs treated with IGF-1 (10 nM) or vehicle (upper) and statistical analysis of densitometric measurement of nuclear KLF15 (lower) are shown (*p<0.05 vs. vehicle, n=3). (f) Immunoblots of cell lysates from NRCMs treated with IGF1 (10 nM) or vehicle (left) and statistical analyses of densitometric measurements of BCAT2, BCKDHA and PP2Cm (right) are shown (*p<0.05 vs. vehicle, n=3). (g) WT mice were subjected to 18 hour fasting and then injected with vehicle or insulin (1 Unit per kg). Heart tissues were collected 2 hours after injection. qRT-PCR analysis of indicated genes were examined. The expression was normalized to 18S rRNA and reported as fold change over vehicle (*p<0.05 vs. vehicle, n=5).

2. The link between increased intracellular glucose promoting cell growth through a KLF15-BCAA mechanism needs to be solidified. Does overexpression of GLUT1 or high glucose medium sufficient to induce hypertrophy under basal conditions or after PE?

We reported that overexpression of Glut1 or high glucose medium did not induce cardiomyocyte hypertrophy in the absence of hypertrophic stimuli (page 11 line 9-10). This is consistent with the observation that Glut1-TG mice have the similar heart weight as WT mice at baseline (Liao et al Circulation 2002). To answer the reviewer's question, we performed new experiments to overexpress Glut1 (OE) in cardiomyocytes followed by vehicle or PE treatment. We found that myocyte hypertrophy was similar between Glut1 OE and LacZ after the treatment (Supplementary Fig. 3h, page 8, line 17-18). Since PE treatment per se increases glucose uptake, we speculate that any further increase of glucose uptake by overexpressing Glut1 may not have an additive effect.

(Supplementary Fig. 3h) NRCMs transduced with indicated adenovirus were treated with phenylephrine (PE, 100 μ M) or vehicle for 48 hours. Myocytes were fixed and stained with anti-Troponin T. Cell surface area in each group was quantified and expressed relative to the control (* p <0.05 vs. LacZ/Vehicle, n =3).

3. BCAA accumulation in the heart due to its reduced degradation appears to be detrimental to the heart, as suggested by Sun's 2016 study. In current study, KLF15's rescue is dependent on PP2Cm-BCAA degradation pathway also suggested a detrimental role of BCAA accumulation. Furthermore, KLF15 is known to be reduced in many heart disease states. This leads the reviewer to wonder why/how the Glut1-TG heart, which exhibited reduction in both KLF15 and BCAA degradation pathway, was healthy at baseline and protected from TAC? The authors should discuss.

The reviewer raised a critical point. We believe that the seemingly contradictory outcome observed in Glut1-TG hearts reflects the complex role of substrate metabolism in modulating myocardial energetics and signaling transduction. During the development of heart failure, downregulation of fatty acid oxidation (FAO) compromises myocardial energy supply while increased glucose utilization through endogenous mechanisms is not sufficient to maintain myocardial energetics. Overexpression of Glut1, an insulin-independent glucose transporter, however, increased cardiac glucose uptake beyond the level of insulin stimulated glucose uptake. This tremendous capacity of glucose supply allows glucose to become the predominant fuel under conditions of impaired FAO (Liao et al, Circulation 2002, Luptak et al, Circulation 2005). Thus, Glut1-TG hearts maintained normal myocardial energetics after TAC and delayed the progression to heart failure. Interestingly, compared to other measures of sustaining myocardial energetics, such as preventing the downregulation of FAO, improved function in Glut1-TG after TAC was not accompanied by the expected reduction of cardiac hypertrophy (Kolwicz SC Jr et al. *Cir Res*, 2012). These observations are consistent with the role of glucose in regulating cell growth independent of its role in ATP production, as shown in the present study.

The reviewer is also concerned that downregulation of BCAA degradation pathway in Glut1-TG did not result in negative outcome after TAC as described in the PP2Cm KO hearts which are defective in BCAA catabolism. According to the study by Sun et al. Circulation 2016, BCKAs accumulation in PP2Cm after TAC contributes to the development of cardiac dysfunction. This is because PP2Cm directly activates the enzyme that metabolize BCKA. However, in Glut1-TG heart, the entire BCAA degradation pathway was downregulated including the enzyme converting BCAAs to BCKAs, we did not observe BCKAs accumulation in Glut1-TG heart (Fig. 1f). In addition, chronic accumulation of BCAA impairs glucose metabolism (Li et al. Cell Metabolism 2017), which is rescued by overexpressing Glut1 in the heart. Thus, the metabolic consequences of decreased BCAA catabolism in Glut-TG heart are likely compensated by increased glucose uptake.

These points are added to the manuscript (page 16, line 16-25, page 17, line 1-12).

4. “Suppression of BCAA Degradation by Glucose Is Required for Maintaining mTOR Activity during Cardiac Hypertrophy”. The claim is not well supported. The authors showed PE-induced mTOR activation requires glucose or BCAA accumulation. But there is no data supporting intracellular glucose level and BCAA accumulation is connected/related after PE treatment or PE needs both to activate mTOR. In addition, it has been reported that BCAA, such as Leu, could directly activate mTOR.

We apologize that this key point was not clear to the reviewer. The evidence for the two questions are: 1) If we knockdown glut1 or remove glucose in the culture medium, PE induced downregulation of BCAA degradation enzymes is abrogated (Fig. 3b; Supplementary Fig. 4l). These observations show that glucose level and BCAA accumulation is connected/related after PE treatment. 2) The removal of glucose prevents PE-induced mTOR activation (Fig. 4b; Supplementary Fig. 4a). Similarly, restoration of BCAA degradation by KLF15 overexpression is also sufficient to prevent mTOR activation in response to PE treatment (Fig. 4e). These observations show that PE needs both to activate mTOR. We agree with the reviewer that BCAAs could directly activate mTOR, and have discussed this point in the manuscript (page 14 line 12-14, line 20-21). The present study is to provide new mechanisms on how metabolic signaling leads to increased intracellular BCAA level under growth conditions.

REVIEWERS' COMMENTS:

Reviewer #1 (Remarks to the Author):

The revision has addressed key concerns to the original report.

Reviewer #2 (Remarks to the Author)

(No further comments from this reviewer)

Reviewer #3 (Remarks to the Author):

The authors have responded well to my queries.